# Targeting pleuro-alveolar junctions reverses lung fibrosis in mice

Adrian Fischer[1,2,19], Wei Han[2,3,4,19] ✉, Shaoping Hu[2,3,4,5,19],
Martin Mück-Häusl[2,6,19], Juliane Wannemacher[2,3,19], Safwen Kadri[2,6], Yue Lin[2,3],
Ruoxuan Dai[2,3], Simon Christ[2,3], Yiqun Su[2,3], Bikram Dasgupta[2,3],
Aydan Sardogan[2,3], Christoph Deisenhofer[2,3], Subhasree Dutta[2,3], Amal Kadri[2,3],
Tankut Gökhan Güney[2,3], Donovan Correa-Gallegos[7], Christoph H. Mayr[6],
Rudolf Hatz[8], Mircea Gabriel Stoleriu [8], Michael Lindner[8,9],
Anne Hilgendorff[10,11], Heiko Adler [3,12,13], Hans-Günther Machens[14],
Herbert B. Schiller[6,15], Stefanie M. Hauck [16] & Yuval Rinkevich[17,18] ✉

Lung fibrosis development utilizes alveolar macrophages, with mechanisms that are incompletely understood. Here, we fate map connective tissue during mouse lung fibrosis and observe disassembly and transfer of connective tissue macromolecules from pleuro-alveolar junctions (PAJs) into deep lung tissue, to activate fibroblasts and fibrosis. Disassembly and transfer of PAJ macromolecules into deep lung tissue occurs by alveolar macrophages, activating cysteine-type proteolysis on pleural mesothelium. The PAJ niche and the disassembly cascade is active in patient lung biopsies, persists in chronic fibrosis models, and wanes down in acute fibrosis models. Pleural-specific viral therapeutic carrying the cysteine protease inhibitor Cystatin A shuts down PAJ disassembly, reverses fibrosis and regenerates chronic fibrotic lungs. Targeting PAJ disassembly by targeting the pleura may provide a unique therapeutic avenue to treat lung fibrotic diseases.

Half of all deaths in the industrialized world result from fibrosis of lung, heart, liver or kidney. Fibrosis occurs in many chronic diseases and injuries, and it is usually the critical stage that engenders organ dysfunction, failure, and death. The lung is especially vulnerable to fibrotic pathologies, due to its interface with the environment and its delicate intricate structure. Indeed, there are over two-hundred individual interstitial fibrotic lung diseases, including numerous viral, bacterial, genetic and autoimmune diseases, not to mention the fibrotic effects of chronic smoking and environmental toxins[1–4]. SARS-CoV infection also leads to permanent lung fibrosis[4,5].

Pulmonary fibrotic diseases almost invariably lead to interstitial scarring and thickening of the connective tissue beneath the pleural membranes that encase the lung[6–12]. These regions, termed as pleuro-alveolar junctions (PAJs), are where the pleural membranes interface with the alveolar interstitium. They are hypothesized to serve as sites of fibrotic remodeling that contribute to disease progression. The gradual build-up of extracellular matrix (ECM) within deep lung tissue, stiffens the lungs, which impairs gas exchange and causes progressive respiratory failure[13,14].

The exact involvement of the PAJ and its connective tissue in fibrosis development remains unclear, as well as how this microenvironment interconnects with inflammation to initiate and sustain fibrosis[15–20]. Inflammation is necessary to pathologic healing with macrophages playing a key role in modulating pulmonary fibrosis[21] and profibrotic signaling events[22,23]. In idiopathic pulmonary fibrosis (IPF), macrophages exhibit diverse roles and distinct phenotypes depending on their location and activation state within the lung microenvironment. Macrophages can be classified based on location, origin, and immune phenotypes, including alveolar macrophages (AMs) and interstitial macrophages (IMs), monocyte-derived inflammatory macrophages (MDMs) and tissue-resident macrophages[24,25]. Recent scRNAseq studies of human lung fibrosis have identified two

categories of AMs: pro-inflammatory and pro-fibrotic[20,26,27].These macrophage subsets contribute to fibrosis by secreting cytokines, chemokines, and reactive oxygen species, which in turn promote fibroblast activation, collagen deposition, and tissue remodeling within the alveolar interstitium. Notably, AMs have been identified as significant contributors to the increased levels of Interleukin 18 (IL-18) and Interleukin 18 receptor alpha chain (IL-18Rα) in the lungs of individuals with IPF[28]. IL-18-expressing AMs are the major immune cell population in human IPF. IL-18 knockout mice are resistant to bleomycin-induced pulmonary fibrosis and elements of IL-18 signaling are elevated in serum of IPF patients[28,29]. A comprehensive understanding of the interplay between AMs, PAJs, and pulmonary fibrosis remains elusive. This knowledge gap is of paramount importance, given that debilitating lung conditions such as IPF are virtually untreatable, with existing therapies merely providing temporary relief over a limited timeframe[30].

In this study, we reveal a new pathological axis for AMs in PAJs that sustains chronic interstitial scarring and fibrosis. Using novel genetic and chemical techniques to fate map pre-made lung ECM in animal models of lung fibrosis, we uncover a novel mechanism of connective tissue proteolysis at PAJs with subsequent mobilization of pleural macromolecules inwards that feeds and sustains pulmonary fibrosis. The fibrosis cascade is initiated by AMs communicating with lining mesothelium through IL18R and Cathepsin B proteolysis on PAJs. Targeting PAJs disassembly not only prevents fibrosis from initiating but when given therapeutically resolves and reverses fully advanced fibrosis in chronic animal models.

## Results

### Injured lungs translocate pleural ECM inwards to activate myofibroblast activity

Damaged and inflamed lungs accrue connective tissue, the provenance of which is incompletely understood. As our model system visualized the sources of connective tissue, we locally tagged and fate-mapped the ECM underneath the mesothelial lining of the lungs, termed here PAJ. By using N-hydroxysuccinimide ester fluorescein isothiocyanate (NHS-FITC) intrapleurally to tag the matrix we could follow any matrix transfer in real time in live mice (Fig. 1a, b and Supplementary Video 1). NHS esters bind to $NH_2$ groups of amino acids covalently and irreversibly, excluding any possibility of leakage[31]. We confirmed absence of any leakage in vivo (Fig. 1b and Supplementary Fig. 1a).

Second-harmonic generation (SHG) microscopy has become a valuable method for examine fibrillar collagen within diverse tissue contexts. Leveraging its physical principles, SHG microscopy offers high sensitivity to the structural nuances of collagen fibrils and fibers, making it particularly adept at detecting pathological changes characteristic of conditions such as cancer, fibrosis, and connective tissue disorders[32,33]. We found that labeled NHS-FITC+ pleura matrix coincided with SHG (Supplementary Fig. 1b). The SHG overlapped with mature NHS-FITC+ collagen fibers in lung pleura, but also with additional structurally immature matrix that filled gaps/spaces between adjacent mature collagen fibers. This more complete profile of matrix is undetected by SHG signal alone, confirming the utility of our new method to track matrix (Supplementary Fig. 1b). We also confirmed that interstitial fibroblasts, immune cells and mesothelial cells were unlabeled with NHS-FITC in our system (Supplementary Fig. 1c). Supplementary Video 2 captures a 3D view of the more complete profile of pleural matrix including structurally mature (NHS+ SHG+) and immature macromolecules (NHS+SHG−). These findings indicate lung surfaces at PAJs are composed of mature and immature connective tissue macromolecules, that is invisible to classic SHG microscopy.

To test the mobility of the immature matrix in response to disease in mice, we applied bleomycin oropharyngeally (Fig. 1c). We chose bleomycin because it is used as a model for pulmonary fibrosis and is characterized in depth[34]. Oropharyngeal instilment of bleomycin in

animals causes acute injury to the lung epithelium, leading to inflammation (first 1–9 days), and extensive pulmonary scarring (peaks around day 14), within the airways and interstitium leading to respiratory failure. Fibrosis is subsequently resolved from three weeks post-bleomycin instilment and onwards[35–37]. It thus allowed us to study the dynamics of matrix during three key steps in lung disease: inflammation, fibrosis and fibrosis resolution. Strikingly, bleomycin induced extensive inward transfer of NHS-FITC+-matrix from PAJs into airways and interstitium. This transfer of NHS-FITC+-pleural matrix was progressive, starting before day 10, peaking at day 14 and thereafter subsiding (Fig. 1d). NHS-FITC+-matrix entered the lower lobes first (Fig. 1d), irrigating the outermost airways and their surrounding interstitial space. Over subsequent days, NHS-FITC+-matrix thickened and solidified within deep lung tissue, forming fibrotic foci with NHS-FITC+ that were picrius red-positive and methyl blue-positive for collagen fibers (Fig. 1d, e). NHS-FITC+ also accumulated around the bronchi and bronchioles (Fig. 1f and Supplementary Video 3). We did not detect the transfer of NHS-FITC+ matrix in other organs (i.e., liver) from bleomycin-treated animals, nor in labeled lungs without bleomycin-treatment, excluding random diffusion of NHS-FITC+-matrix in the animals (Supplementary Fig. 1d). Crucially, large patchy foci of PDGFRα+ myofibroblasts formed at sites were NHS-FITC+-matrix had accrued (Fig. 1g). Histology and cell quantification of bleomycin-treated lungs indicated that the tissue distribution of NHS-FITC+-matrix within the lung interstitium coincided with fibroblast activation (Fig. 1h). Our data thus indicates that inward transfer of NHS-FITC+-matrix is spatially and temporally linked to myofibroblast activation. High resolution 3D imaging of the complete protein distribution on lung surfaces showed clear reduction of matrix (NHS-FITC signal), associated with structural changes, in response to bleomycin treatment (Supplementary Fig. 1e and Video 4). This reduction of NHS-FITC + matrix from surfaces of diseased lungs directly correlated with loss of fine fibrillar volumes and with the extent of inward transfer of NHS-FITC+ matrix, none of which occurred in healthy pleura (Supplementary Fig. 1f).

Next, we sought to define the protein constituents of the transferred matrix by combining pleural matrix fate-mapping with mass-spectrometry of deep lung tissue. Briefly, we tagged the matrix on PAJs with a Biotin-conjugated EZ-link sulfo-N-hydroxysuccinimide ester. We followed up by subjecting mice to bleomycin-induced injury (Supplementary Fig. 2a). Two weeks post-injury, we collected diseased lungs, separated pleura from deep lung tissue, and purified labeled matrix specifically from deep lung tissue via Streptavidin pull-down followed by mass-spectrometric proteomics of all tagged peptides (see "Methods"). The proteomics analysis revealed that 52% of all transferred ECM proteins belong to ECM glycoproteins, with collagens accounting for 36%, ECM-affiliated proteins for 8%, and proteoglycans for 4% (Supplementary Fig. 2b). Notably, the identified collagens include members from both the fibrillary (Col1a1, Col1a2, Col3a1) and basement membrane (Col4a1, Col4a2) collagen families (Supplementary Fig. 2c). GO biological process analysis indicated enrichment primarily in ECM organization, extracellular structure organization, and external encapsulating structure organization. In terms of cellular components, significant enrichment was observed in collagen-containing ECM, basement membrane, and endoplasmic reticulum lumen. GO molecular function analysis showed enrichment in platelet-derived growth factor binding, protease binding, and phospholipase inhibitor activity. Phenotype analysis suggested that the transferred matrix resembles scar tissue in humans (Supplementary Fig. 2d). Overall, these data uncover inward transfer of pleural matrix, deep into lung tissue, occurring over days, effectively laying down connective tissue macromolecules that correlates with picrius red+ methyl blue+-collagen fibers and with fibroblast activity in injured lungs (Fig. 1i).

The premade transferred ECM being a novel concept, we next sought to investigate its relative contribution compared to the

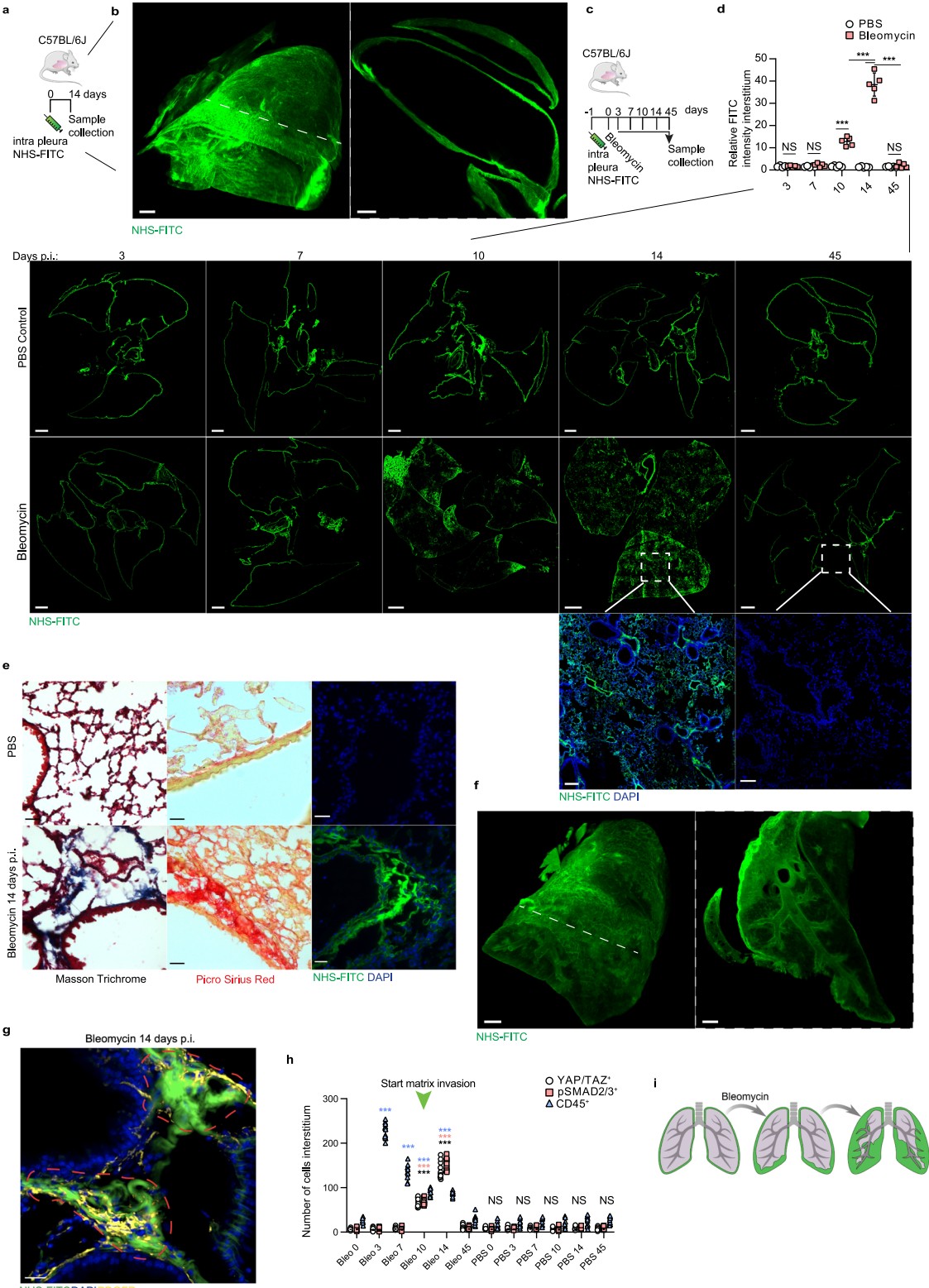

traditionally recognized newly synthesized ECM. To do this, we employed in parallel to NHS fate mapping, a second methodology, specifically designed to visualize and quantify de novo deposited ECM by administering non-canonical amino acids (ncAAS). Systemic administration of ncAAS in mice led to integration of ncAAS into all newly synthesized proteins[38], and enabled subsequent visualization and quantification of newly synthesized ncAAS⁺ proteins (Fig. 2a and see "Methods"). Wild-type mice were daily given ncAAS

intraperitoneally for three days, revealing a gradual and significant cytoplasmic accumulation of de novo synthesized proteins within many cells and regions of the lung, but without detectable extracellular contributions (Fig. 2b). We detected no significant overlap between regions of ncAAS⁺ proteins (matrix synthesis) and NHS-FITC⁺ pleural matrix (Fig. 2b), confirming that matrix transfer (NHS-FITC⁺) and de novo matrix synthesis (ncAAS⁺) are parallel processes, the extent of which can be analyzed simultaneously. We therefore applied

**Fig. 1 | Extracellular matrix -fate mapping reveals inward matrix transfer during lung injury. a** Schematic representation of the pleural matrix fate mapping setup. C57BL/6J mice were intrapleurally injected with NHS-FITC labelling mix and two weeks later lungs were harvested. **b** Representative light sheet images of murine lungs two weeks post-intrapleural NHS-FITC injection. $n = 6$ biological replicates (C57BL/6J WT mice) and three independent experiments. Scale bars: 500 μm. **c** Schematic representation of bleomycin pleural matrix fate mapping setup. Mice were intrapleurally injected with NHS-FITC labelling mix. The next day bleomycin was applied and lungs were collected after indicated timepoints. **d** Representative histology images of mouse lungs 3-, 7-, 10-, 14- and 45-days post-bleomycin installation (p.b.i.). $n = 6$ biological replicates (C57BL/6J WT mice) and 5 independent experiments. The graphic above shows relative FITC intensity in the interstitium in control and treated mice at the various timepoints. Data represented are mean ± SD. A two-sided independent T-test was used for the comparison of two groups (Day10: $p = 0.00013$, Day14: $p = 8.61e-05$)(***$P < 0.001$; NS= not significant). Scale bars: 500 μm (Overview); 100 μm (Highlight). **e** Representative images of

mouse lung interstitum 14 days p.b.i. Collagen fibers are stained blue (Masson trichrome) and red (Pico Sirius Red). $n = 6$ biological replicates (C57BL/6J WT mice) and 6 independent experiments. Scale bars: 50 μm. **f** Representative light sheet microscopy images of murine lungs two-weeks p.b.i. $n = 6$ biological replicates (C57BL/6J WT mice) and 6 independent experiments. Scale bars: whole organ 500 μm. **g** Representative histology images of mouse lungs 14 days p.b.i. Mice were intrapleurally injected with NHS-FITC labelling mix. The next day bleomycin was installed. Regions with NHS-FITC+ and PDGFRα+ signal are highlighted in red. $n = 6$ biological replicates (C57BL/6 J WT mice) and 6 independent experiments. Scale bars: 50 μm. **h** Quantification of inflammatory cells (CD45+) and fibroblasts (pSMAD2/3+ and YAPTAZ+) in mouse lungs post-oropharyngeal bleomycin or PBS installation. $n = 6$ biological replicates (C57BL/6J WT mice) and 6 independent experiments. Data represented are mean ± SD. One-way ANOVA was used for the multiple comparison (NS= not significant). **i** Schematic representation of bleomycin induced matrix invasion. Upon oropharyngeal bleomycin installation pleural matrix pools invade lung interstitium.

matrix synthesis (ncAAS) and matrix transfer (NHS-FITC) protocols in the presence of bleomycin instillation (Fig. 2c). We then collected the fibrotic lungs for histochemical analysis and determined the degree of inbound transferred collagen (NHS-FITC+) and newly synthesized (ncAAS+) collagen within diseased lung samples.

Lungs on day 10 post bleomycin showed newly synthesized collagens in the cytoplasm of many cells but with negligible newly synthesized collagens in the extracellular space. Strikingly, however, there were already significant amounts of inbound transferred NHS-FITC+ collagens within the lung extracellular space at this time (Fig. 2d). This clearly indicates that matrix is transferred from PAJs before any newly synthesized matrix is deposited. On day 14 lungs had significant amounts of newly synthesized collagen (ncAAS+) within the extracellular space. This newly synthesized collagen already made up about a third of the total collagen. The proportion of imported premade collagen (NHS-FITC+) rose to almost two-thirds on 14 days post bleomycin in fibrotic lungs (Fig. 2d). Our dual in vivo ECM tracing methods indicate that significant deposition of ECM begins only after transfer of pleural matrix, and that transfer of pre-made ECM constitutes a part of the fibrotic material.

To test if the transfer of pre-made ECM is truly a general facet of fibrosis, we opted for a viral-based mouse model in which intra-nasal herpes virus is instilled into IFN-γ-R$^{-/-}$ mice leading to pneumonia and systemic inflammation, followed by pulmonary fibrosis[39,40]. Whereas chemical bleomycin models undergo fibrosis resolution post-day 28, the herpes virus instilment models remain chronically fibrotic and symptoms do not resolve. We followed our 'tagging' protocol of pleural matrix (NHS-FITC) with herpes virus instilment to induce fibrosis. Lungs on days 15 post herpes virus had widespread transferred matrix (NHS-FITC+) in deep lung tissue, accompanied by fibroblast activation at those sites (Supplementary Fig. 3a), similarly to chemical-induced fibrosis. Strikingly, inbound transferred matrix overlapped precisely in time and space with fibrosis in the two dichotomous chemical and viral models of lung fibrosis (Supplementary Fig. 3b). Bleomycin instilled lungs showed negligible inbound transferred ECM on day 45, consistent with fibrosis resolution. In contradistinction, viral injured lungs still had copious amounts of transferred NHS-FITC+ fibrous material within deep lung tissue on day 45, and structural fibrosis.

To further verify our findings in a separate induction system, we over-expressed TGFβ in lining mesothelium in animals, the best-studied mediator of fibrotic processes[41]. A single injection of recombinant TGFβ into the intrapleural space provided robust phosphorylation of SMAD in pleural lining mesothelium (Supplementary Fig. 4a). One week after TGFβ application, a thickening of the pleural matrix pool was detected without significant transfer. However, after 2 weeks, the lung was completely filled with pre-made and transferred matrix, which was accompanied by increased TGFβ signaling,

indicating of fibrotic tissue (Supplementary Fig. 4b, c). This suggests a stepwise process by first a mesothelial TGFβ induced accumulation of pleural matrix and then a release or activation process that initiates matrix transfer. To be sure that lining mesothelium is responsible for our observed effects, we reverted to our AAV system. This time we used a vector encoding an activated variant of TGFβ, application of the vector resulted in stable expression of TGFβ in mesothelial linings (Supplementary Fig. 4d). Over-expression of mesothelial TGFβ resulted in massive influx of pre-made ECM, activation of pSMADs and increased amounts of M6A+ cells in lung interstitium (Supplementary Fig. 4e, f). The next step was to investigate the effects of mesothelial TGFβ signaling reduction on fibrotic processes. Our strategy employed over expression of a dominant negative form of the TGFβ receptor dead mutant (DN-TGFβRII)[42]. Application of the DN-TGFβRII vector showed a stable transduction and expression of the receptor in the mesothelial lining of the visceral pleura (Supplementary Fig. 4g). Next, we investigated the effects of mesothelial TGFβ inhibition on bleomycin-induced fibrosis (Supplementary Fig. 4h) by instilling bleomycin in these animals. Indeed, inhibition of mesothelial TGFβ signaling blocked transfer of pre-made ECM, blocked accumulation of phosphorylated SMAD and bleomycin-induced mortality (Supplementary Fig. 4i, j).

Overall, our fate-mapping in three lung fibrosis models uncover an inward axis of ECM transfer, from PAJs into deep injured lung tissue over days, effectively irrigating lungs with connective tissue that correlates with myofibroblast activation in a temporal and spatial manner (Fig. 1i).

## Transferred ECM originates from pleural mesothelium

As the matrix that moves into lungs is located directly below the pleural mesothelium, we next sought to formally demonstrate if the transferred matrix indeed comes from mesothelial cells, which have been reported to play an important role in pulmonary fibrosis[43,44]. To this end, we used an AAV8-based system, to transduce a collagen 1-FLAG (Col1-FLAG), a collagen 2- FLAG (Col2-FLAG), and a collagen-helix binding CNA35-mCherry reporter tag specifically into mesothelium. Transduced cells that de novo synthesize collagen transcripts, generate a Col1-FLAG, Col2-FLAG or a CNA35-mCherry fusion protein that, once incorporated into new collagen helices, reveals the cell/tissue sources for any newly deposited collagen (Fig. 3a). Intrapleural administration of AAV8-Col1/2-FLAG reporter resulted in a specific viral transduction of the mesothelial lining, without any interstitial labeling (Fig. 3b).

Next, pleural surfaces were additionally labeled with NHS-FITC, to tag pre-existing ECM as above, followed by administration of bleomycin in trachea (Fig. 3c). Injury resulted in transfer of NHS-FITC+-matrix into the interstitium, as well as mesothelial synthesized

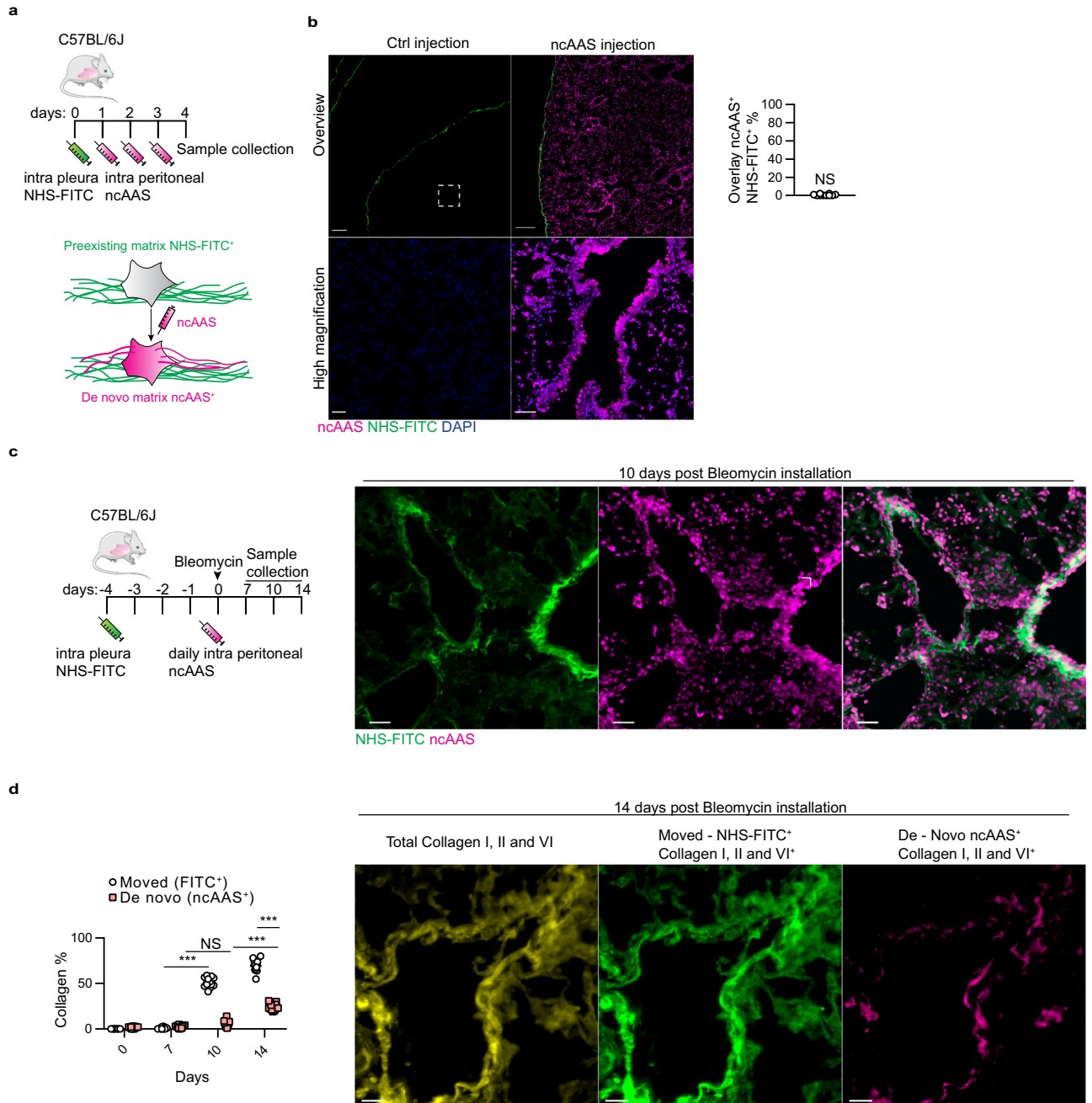

**Fig. 2 | Significant interstitial collagens in fibrotic mouse lungs come from distant sites. a** Schematic representation of non-canonical amino acid (ncAAS) protein tracing setup. C57BL/6J mice were intrapleurally injected with NHS-FITC labelling mix and for three consecutive days, ncAAS was injected intra-peritoneally. Lungs were collected four days post-NHS-FITC injection. ncAAS are randomly integrated in newly synthesized proteins and can be detected with a fluorescence probe using Click-iT chemistry. **b** Representative images of mouse lungs after three days of consecutive ncAAS injections. ncAAS were visualized using Click-iT chemistry. $n = 6$ biological replicates (C57BL/6J WT mice) and 6 independent experiments. Scale bars: 500 μm (Overview); 50 μm (high magnification). Data represented are mean ± SD. A two-sided independent T-test was used for the comparison of two groups (NS= not significant). **c** Representative images of mouse lungs 10 days post-NHS-FITC labeling with daily ncAAS injection. $n = 6$ biological replicates (C57BL/6J WT mice) and 6 independent experiments. Scale bars: 50 μm. **d** Representative images of mouse lungs intrapleurally labeled with NHS-FITC, 14 days p.b.i. ncAAS were injected daily and visualized using Click-iT chemistry. $n = 6$ biological replicates (C57BL/6J WT mice) and 6 independent experiments. Scale bars: 100 μm (left) and 50 μm (right). Data represented are mean ± SD. One-way ANOVA was used for the multiple comparison (*** $P < 0.001$; NS= not significant).

collagen. These NHS-FITC⁺ Col-reporter⁺ double-positive patches revealed that pre-existing and newly synthesized ECM from mesothelium are both transferred deep into lung tissue from pleural surfaces (Fig. 3c). Mesothelial-born pools of ECM are thus constantly generated de novo at PAJs and transferred inwards to fuel lung fibrosis (Fig. 3d).

Having demonstrated in three separate in vivo systems that pleural surfaces transfer ECM into lung fibrosis, we next examined mesothelial gene expression kinetics during bleomycin-induced pulmonary fibrosis. The scRNA-Seq analysis was performed to capture the gene expression dynamics in mesothelial cells, providing insights into their potential role in ECM remodeling during fibrosis, complementing

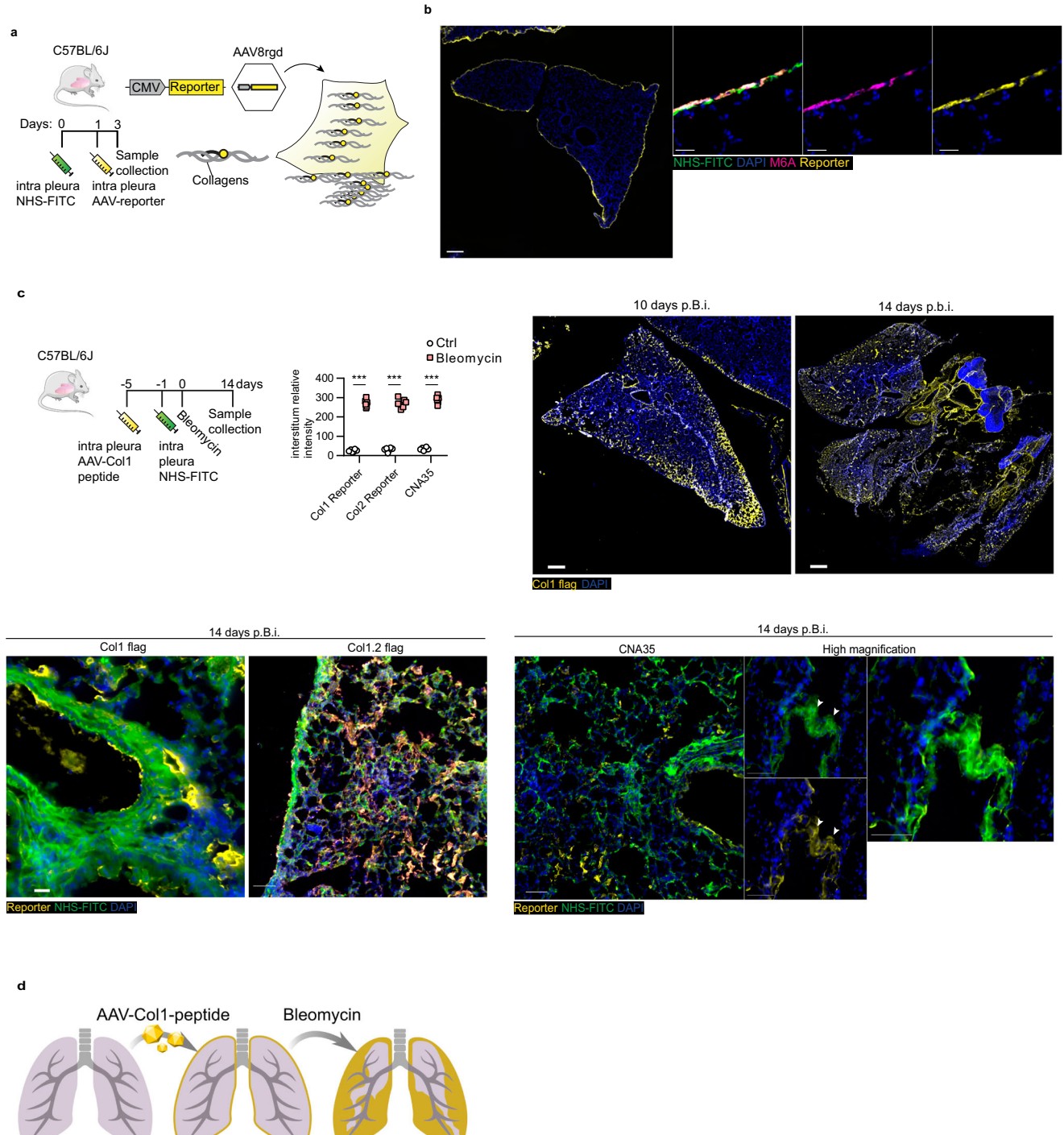

**Fig. 3 | Lung mesothelium is the source of transferred matrix in injury.**
**a** Schematic representation of genetic pleural collagen protein fate mapping in animals. Mesothelial cells were transduced by intra-pleural injection of AAV particles expressing collagen reporters that monitor mesothelial-specific deposition of collagen. **b** Representative histology images of mouse lungs three days post-Col1-reporter injection. $n = 6$ biological (C57BL/6J WT mice) and 6 independent experiments replicates per reporter construct. Scale bars: 500 μm (Overview); 50 μm (high magnification). **c** Schematic, graphic and representative histology images of mouse lungs 10 and 14 days p.b.i. $n = 6$ biological (C57BL/6J WT mice) and

6 independent experiments replicates per reporter construct. Scale bars: 1000 μm (Overview); 50 μm (high magnification). Data represented are mean ± SD. A two-sided independent T-test was used for the comparison of two groups (Col1 reporter: $p = 1.09e\text{-}07$, Col2 reporter: $p = 1.87e\text{-}07$, CNA35: $p = 4.49e\text{-}08$) (***$P < 0.001$). **d** Schematic representation of the mesothelial-specific fate mapping of collagen protein in bleomycin-induced lung fibrosis. Upon oropharyngeal bleomycin installation pleural matrix pools that include newly deposited collagen (yellow) invade lung interstitium.

the observations from ncAAS tagging. We analyzed scRNA-Seq data sets of lung mesothelium from bleomycin treatment on day 0, 3, 7, 10, 14, 21 and 28 post bleomycin installation (Supplementary Fig. 5a) using the Splines and Imtest pipelines[35]. In response to bleomycin instilment,

mesothelial cells synthesized distinct fibrotic ECM proteins over the studied stages. Healthy lung mesothelium (day 0) expressed collagens Col4a3, Col4a4, laminins and mucin family members, consistent with basement membrane maintenance and a lubricating non-fibrosis role

for healthy mesothelium. Three days post-bleomycin, mesothelial cells expressed various fibrillary collagens of type 1, 4, 5, 7, 12, and 14, indicating diverse fibrillary collagens accumulate in pleural surfaces during early stages of fibrosis. On day 7 post bleomycin, mesothelial cells expressed additional new collagen family sub-members of type 6, 7, 14 and 27. Secreted glycoproteins also become incorporated into extracellular protein fibers and were transferred inward into deep lung tissue. On day 10 (Supplementary Fig. 5a), thiol proteases such as Cathepsin family members B, C, D, H, and S had peaked in expression. At more progressive stages post-bleomycin, i.e., on day 28, mesothelial cells decreased collagen and thiol protease expression levels, reverting ECM protein expression back to that seen in baseline homeostatic states. These data demonstrate that the pleural lining dynamically upregulates its ECM expression and degradation profiles during injury, consistent with progression of ECM transfer and lung fibrosis.

## Diseased human lungs undergo vigorous inward transfer of pleural matrix

To study if human diseased lungs undergo ECM transfer in the same way as in our three mouse models, we adapted our NHS-FITC technique to human peritumor and IPF lung samples (Fig. 4a). Human lung PAJ (Supplementary Video 5) has the same two types of connective tissue organization as mouse lung surfaces, i.e., reticular frames of thick collagenous SHG$^+$-NHS-FITC$^+$ fibers that are bathed in SHG$^-$-NHS-FITC$^+$ macromolecules. As in mouse lungs, labeled ECM on human lung surfaces transferred into the interior of diseased human lung tissue (Fig. 4b and Supplementary Video 5). Two-photon images of surface-labeled diseased human lung biopsies revealed accumulation of NHS-FITC$^+$-fibers in deep alveolar and interstitial spaces and also around the bronchioles and blood vessels (Fig. 4c and Supplementary Video 6). These data demonstrate that fibrotic sources from transferred ECM remains chronically active in fibrotic and diseased human lungs.

To study the composition of transferred ECM within human lungs, we tagged diseased biopsy pleura ECM with NHS-EZ-link before incubation. We then separated pleura from deep interstitial tissues and extracted the transferred ECM from deep interstitial tissues using our NHS-EZ-link$^+$ system, followed by mass spectrometry proteomics (Fig. 4d). Supplementary Data 2 shows the full inventory list of human extracellular macromolecules that have transferred. Among all transferred ECM proteins, ~30% are ECM glycoproteins, 22.28% are ECM regulators, and 15.84% are collagens (Fig. 4e). GO term analysis revealed significant enrichment in biological processes such as ECM organization, external encapsulating structure organization, and extracellular structure organization. In terms of cellular components, the most enriched categories were collagen-containing ECM, endoplasmic reticulum lumen, and intracellular organelle lumen. Molecular function analysis highlighted enrichment in endopeptidase inhibitor activity, protease binding, and serine-type endopeptidase inhibitor activity. Phenotype analysis indicated that the NHS-EZ-link+ protein matrix resembled atrophic scars in humans (Fig. 4f).

In agreement with our multi-photon imaging, we detected abundant fibrillar collagenous fibers such as Collagen type I and III, and their covalent crosslinking enzymes such as Lysyl oxidase (Lox) and Transglutaminases (Tgm) within the human transferred pool. Lox and Tgm are both involved in connective tissue remodeling and maturation and activation of fibroblasts into pathologic myofibroblasts (Fig. 4g). We also found proteins involved in basement membrane formation and stability, such as Collagen type IV, VI, including elastic fiber complexes such as elastins and fibrillins: needed to maintain lung pliancy and elasticity, all of which were translocated from PAJs. Overall, we identified a human protein inventory that contributes to tissue rigidity in multiple ways.

By analyzing the relative fractions of NHS-EZ-link$^+$-proteins remaining on lung surfaces versus those transferred, we found that individual proteins had distinct translocation profiles (Supplementary Fig. 6a). For example, elastic fibers and fibrillary collagens had a high translocation index of -1.2, whereas basement membrane components were notably slow, with a poor index of -0.8 (Supplementary Fig. 6b). This implies matrix transfer affects organ stiffness more rapidly than reduction of surface elasticity. Apolipoprotein (LPA) also stood out as having an particularly high translocation index, consistent with LPA serving as an early fibrosis biomarker[45]. These widely varying translocation profiles indicate that protein liberation from pleural surfaces is dynamic and that the protein composition that bathes injured lungs changes its constituent proteins depending on their individual rates of movement.

## Proteolysis of lung surfaces by alveolar macrophages through IL-18/IL-18 receptor

Having established that pleural mesothelium sends pre-formed ECM inwards, we sought to establish the initiation signals within the outer pleura that detect alveolar injury. Cell-cell communication based on ligand-receptor analysis of scRNAseq data from bleomycin-instilled lung fibrosis models, revealed the most potent communication in the pleura during fibrosis occurred between monocyte-derived AMs (Supplementary Fig. 5b) and mesothelial cells, through IL-18R on mesothelial cells and IL-18 on AMs (Supplementary Fig. 5c). We confirmed IL-18R protein expression on pleural lung surfaces of both bleomycin-treated mice and human IPF lungs (Fig. 5a). IL-18 protein expression was absent from healthy lungs. However, IL-18 was expressed during bleomycin and viral-induced lung fibrosis in alveolar and interstitial immune cells but absent from mesothelial cells during injury (Fig. 5b, c). Co-immunostainings of IL-18 with CD45 and CD11c proteins (Fig. 5d) confirmed that most IL-18$^+$ cells were indeed AMs. Moreover, culturing lung biopsies with recombinant IL-18 induced matrix transfer from PAJs. We used GM-CSF treated bone marrow derived macrophages (BMDMs), which are known to act as a proinflammatory cytokine that enhances antigen presentation and drives macrophages into a proinflammatory phenotype that produces inflammatory cytokines[46]. We found that transwell co-cultures of lung biopsies with IL-18$^+$ macrophages also induced significant transfer of NHS+ matrix from PAJs. Consistent with the crucial role of IL-18, a blocking antibody against IL-18 prevented this matrix transfer (Fig. 5e, f).

These findings are consistent with studies showing knockout of AMs impedes bleomycin induced pulmonary fibrosis[47,48] and more recently that IL-18$^{high}$-AMs are highly present in human lung disease, particularly in lung fibrosis[26]. Our findings are also consistent with studies showing lungs of IPF patients overexpress IL-18 and that serum levels of IL-18 correlate with advanced pulmonary fibrosis[28].

Having identified the IL-18/IL-18R axis in driving inward transfer of pleural matrix ex vivo and in mouse models, we reanalyzed human lung cell atlas data[26]. Our re-analysis of the scRNAseq data (Supplementary Fig. 7a) showed ~17% of all AMs in human IPF are IL-18$^{high}$ with ~300x increased IL-18 expression compared to all other lung cells and ~10x increased compared to other AMs (Supplementary Fig. 7b and methods). Gene enrichment analysis indicated IL-18$^{high}$ AMs had two-fold higher expression of genes associated with proteolytic, antiviral and antibacterial functions (Supplementary Fig. 7c) indicating a functionally and transcriptionally distinct population of macrophages drives pleural ECM disassembly. Cell-cell communication analysis in human IPF (Supplementary Fig. 7d) showed Resistin, ANXA1 and SIGLEC1, work in an autocrine feedback loop and are both sender and receiving signals in IL-18$^{high}$ AMs. We also observed specific cell-cell communication profiles in IL-18$^{high}$ AMs with B-cells (APRIL), bronchial goblet cells and alveolar epithelial cells (SAA and UGRP1) and mesothelial cells (IL-18R, LIGHT, TWEAK). Taken together our data reveal IL-18$^{high}$ AMs activate IL-18R$^+$ mesothelial cells in human IPF and in mouse

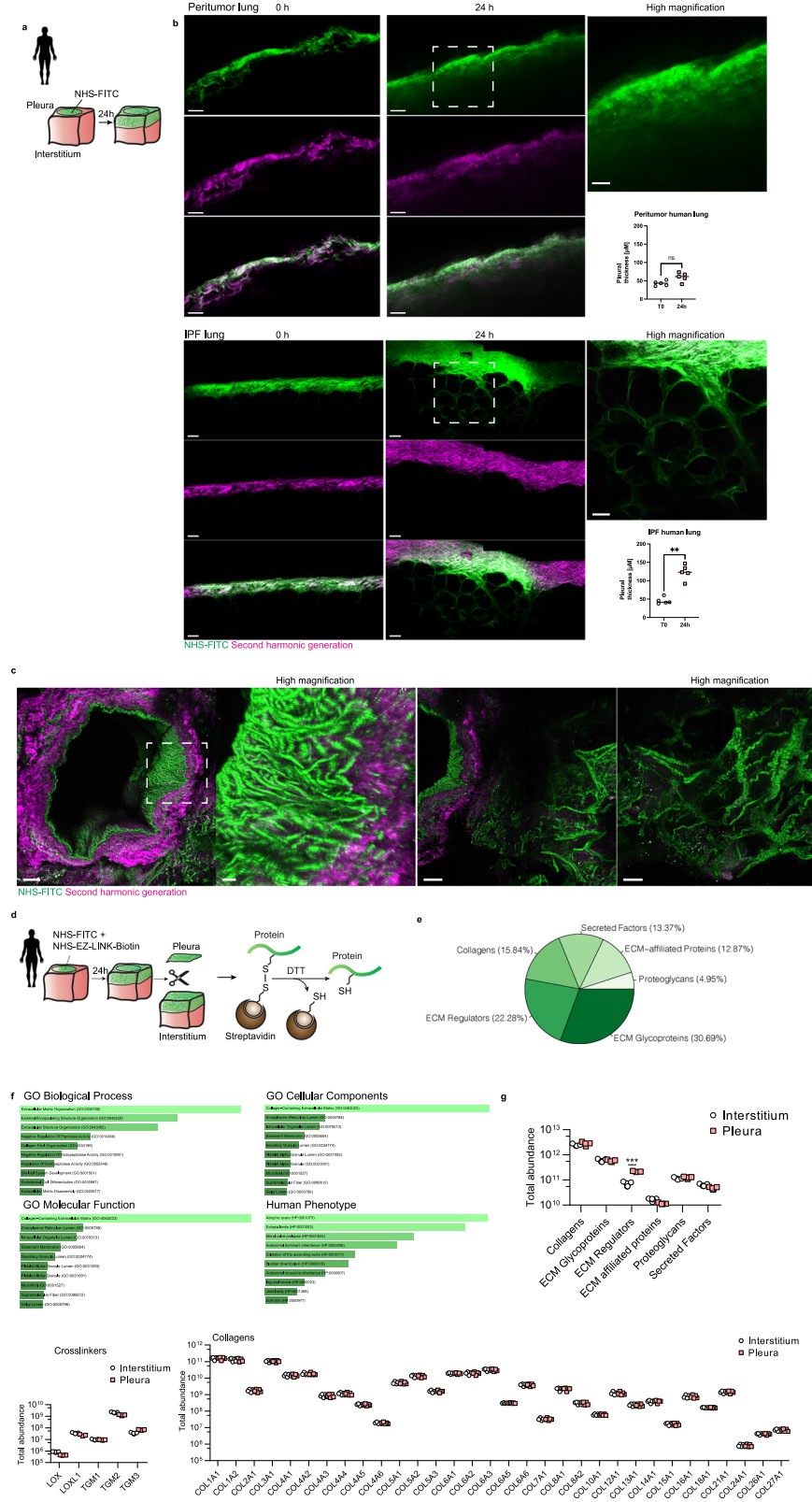

models of lung fibrosis, leading to ECM disassembly at PAJs and transfer into the lung.

## IL-18^high AMs upregulate Cathepsin B on mesothelium to trigger matrix disassembly

Next, we sought to characterize the signaling pathways in mesothelium by which IL-18 protein and IL-18^high AMs trigger matrix transfer and

fibrosis. scRNAseq analysis of human interstitial lung disease patients (Fig. 6a) and of bleomycin lung fibrosis models indicated that thiol proteases are upregulated in mesothelial cells. Specifically, mesothelial cells upregulate cathepsins, and their inhibitory counterparts, the cystatins, in response to bleomycin in animals, and also in fibrotic human lungs. Immunolabeling of our ex vivo tissues treated with recombinant IL-18 and cocultured with IL-18^high AMs upregulated

**Fig. 4 | Human invading matrix resembles scar tissue. a** Schematic representation of human tissue ex vivo assay. Pleural surfaces from human tissue explants where labeled with NHS-FITC and incubated for 24 h. **b** Representative multiphoton images of NHS-FITC labelled in both peritumor and IPF human lung tissues. Scale bars: 25 μm. $n = 5$ biological replicates (peritumor human lung tissue or IPF human lung tissue) and 5 independent experiments. A two-sided independent T-test was used for the comparison of two groups (IPF: $p = 0.0079$)(ns = not significant; **$P < 0.01$). **c** Representative high magnification multiphoton images of NHS-FITC labelled in IPF human lung tissues. $n = 5$ biological replicates (IPF human lung tissue) and 5 independent experiments. Scale bars: 100 μm and 20 μm. **d** Schematic representation of proteomic identification of NHS-EZ-LINK⁺ transferred human matrix components. **e** Identified matrisomal proteins in mass spectrometry analysis. **f** GO enrichments of matrisome proteins identified in mass spectrometry analysis. Moved NHS⁺-ECM shows similarity to atrophic scar tissue. **g** Identified matrisome proteins in mass spectrometry analysis of NHS-EZ-LINK-Biotin⁺ proteins. Showing which NHS + -ECM elements are enriched in pleura and interstitium after 24 h. $n =$ five biological replicates and 5 independent experiments. Data represented are mean ± SD. Single comparison was performed by two-sided independent T-test (ECM regulators: $p = 2.16\text{e-}07$)(***$P < 0.001$).

Cathepsin B on pleural surfaces (Fig. 6b). Cathepsin B expression temporally correlated with matrix transfer in bleomycin models, and Cathepsin B protein expression was absent at 45 days, when fibrosis resolves (Fig. 6c). Moreover, chronic viral fibrosis models sustained Cathepsin B expression in mesothelium on day 45, consistent with a non-resolving model (Fig. 6d). Taken together, our data indicates that IL-18/IL-18^high AMs promote and sustain fibrosis through digestion and liberation of ECM at PAJs.

To test this, mice were injected with bleomycin in trachea, followed by a single pleural injection of a Cathepsin B inhibitor Z-FA-FMK that irreversibly blocks the active center of Cathepsin B. In the presence of bleomycin, pharmacologic inhibition of Cathepsin B completely prevented matrix transfer into lungs, preventing fibrosis development and bleomycin-induced mortality (Fig. 6e, f), which increased survival rates of lung-injured mice to 100%. Crucially, Cathepsin B pharmacological intervention also prevented matrix accrual in human lung biopsies from diseased patients (Supplementary Fig. 8a).

To prove that mesothelial Cathepsin B, specifically, promotes fibrosis, we overexpressed Cathepsin B and Cystatin A (a direct inhibitor that binds and blocks Cathepsin B protease) specifically in the pleural lining. Mesothelial-specific transfection was made possible by pleural injection of a modified AAV8 that expresses RGD peptides on its viral capsid thereby binding to mesothelial cell surfaces (Fig. 6g). Both the Cathepsin B and Cystatin A constructs were specifically expressed in mouse lung mesothelial cells (Fig. 6g). We, therefore, performed pleural injection of NHS-FITC to label ECM at PAJs and treated with bleomycin to induce fibrosis (Fig. 6h). Overexpression of mesothelial Cathepsin B alone, in the absence of bleomycin, led to an increase in NHS-FITC⁺-matrix influx, myofibroblast activation, fibrosis and mortality as compared to control vector-treated animals (Fig. 6h, i). Conversely, suppression of mesothelial Cathepsin B, by overexpressing Cystatin A in mesothelium, blocked ECM transfer from PAJs, even in the presence of bleomycin. Cystatin A-treated animals had no myofibroblast activation, even at the peak of fibrosis on day 14 post bleomycin, and lacked weight loss or mortality (Fig. 6h, i).

Our findings suggested that chronic proteolysis at PAJs and ECM transfer not only activates but also sustains fibrosis. To directly test this idea, we first applied bleomycin to animals and 5 days later administered intrapleural AAV-Cystatin A, tracked the animal's weight and investigated lungs at day 21 post bleomycin instillment (Fig. 6j). Indeed, lung architecture resolved in the Cystatin A-treated group, with significantly less pSMAD2/3+ PDGFRα+ myofibroblasts and absence of weight loss and mortality (Fig. 6k). To further determine the link between matrix transfer and chronic fibrosis, we installed herpes virus in animals. On day 45, with fibrosis already present, we applied intrapleural AAV-Cystatin A followed by lung examination on day 90 (Fig. 6l). Lungs of AAV-Cystatin A treated animals effectively showed reversal of chronic fibrosis, with clear histomorphology resolution and minimal pSMAD2/3-positive myofibroblasts (Fig. 6m).

Taken together, our findings identify a pathomechanism whereby free or AM-bound IL-18 triggers proteolysis at PAJs through Cathepsin B, irrigating lungs with pre-made ECM which further activates myofibroblasts to trigger and perpetuate chronic fibrosis (Fig. 7). Our findings indicate that pharmacological inhibition of Cathepsin B or overexpression of Cystatin A in the pleural mesothelial lining alone may represent an effective treatment to combat chronic organ fibrosis and induce disease reversal. ECM transfer is likely a general facet of tissue/organ injury with potential clinical ramifications to many human fibrotic conditions.

## Discussion

Pulmonary fibrosis is currently incurable. In this study, we reveal a new patho-axis formed by AMs and mesothelium at PAJs. Earlier investigations in transgenic mice have confirmed the indispensable role of AMs in pulmonary fibrosis, alongside the observation of heightened IL-18 levels in cases of pulmonary fibrosis[23,28]. Cathepsin B releases connective tissue macromolecules from PAJs, which chronically irrigate lung interior to sustain fibrosis. Once the Cathepsin B signaling cascade has been activated in the mesothelium the macrophages are no longer needed. Complementarily to this, our experiments also demonstrate that once ECM transfer is blocked at PAJs, interstitial fibroblasts remain inactive even after lung injury and refrain from depositing matrix. Cathepsin B and Cystatin A expression at PAJs are thus a therapeutic vulnerability that could be exploited to prevent or even cure lung disease. Indeed, in both the bleomycin and viral models, we demonstrate that intrapleural mesothelial application of AAV-cystatin induced a significant reduction of active fibroblasts and fibrosis.

Fibrosis had been assumed to be the sole remit of interstitial activated myofibroblasts synthesizing collagen de novo. Our data also show that matrix transfer, together with fibroblast activation, produce the bulk of fibrotic material in fibrotic lungs. We were the first to report import of ECM macromolecules in fibrosis using skin as a model[49] and in abdominal/peritoneal fibrosis[50]. Our findings in lung thus demonstrate macromolecular transfer to be a general feature of tissue/organ healing whereby ECM transfer from the serosal lining likely contributes to fibrotic healing in all organs.

To investigate the initiation of fibrosis in more mechanistic detail, we used -omics approaches and found the IL-18⁺-AM population upregulate genes in pathways combating microbial infections. This indicates that a pulmonary infection leads to permanent activation of AMs at PAJs, therefore amplifying the proteolytic cascade. We speculate that the transferred ECM provides information in forms of altered biomechanical, physical and signaling environments for blood vessels, lung epithelial cells and fibroblasts that in turn furnish extra layers of fibrotic tissue. Our intervention experiments on PAJs revealed that blocking pleural ECM from invading alleviates and reverse fibrosis progression, leading to healthier lung architecture.

In contrast to our previous findings in the fascia of the skin, where fibroblasts moved ECM cargo, the mechanism in the lung involves different cellular and molecular dynamics. Here, we reveal a novel mechanism in lung fibrosis where AMs disassemble and transfer ECM components from PAJs into deeper lung tissues through cysteine-type proteolysis, particularly involving Cathepsin B. This mesothelial activation is facilitated by communication with the AMs via the IL18 receptor. This energetic process indicates that signals at PAJs are

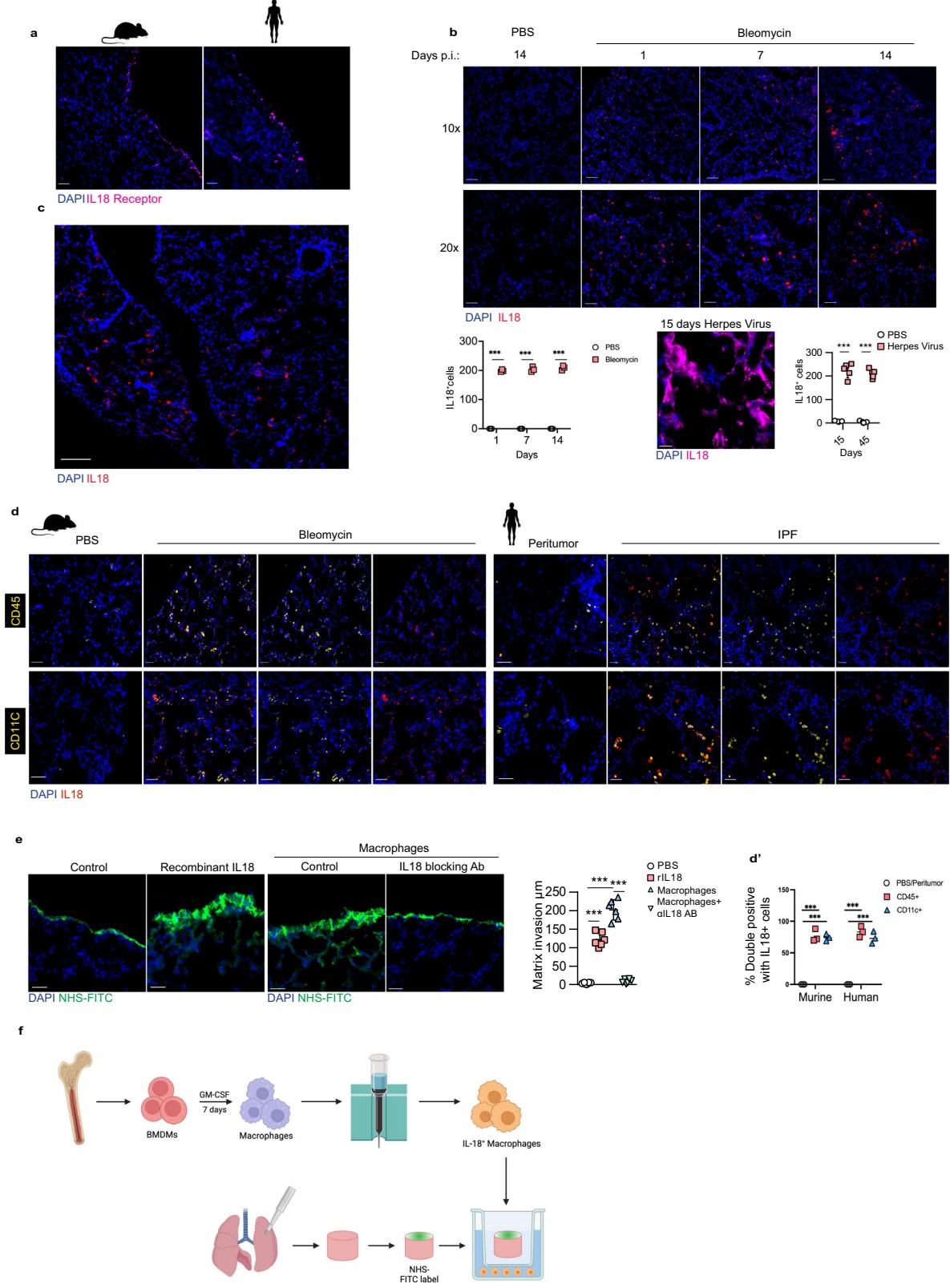

essential for myofibroblast activation and maintains in deep lung tissue.

While our study provides valuable insights into ECM dynamics and protease involvement, we acknowledge certain limitations. Firstly, although we have extensively validated the NHS-ester labeling method for studying ECM in vivo, additional experiments such as Evans blue assays or bleomycin IV administration could further confirm the specificity of the observed matrix transfer. Secondly, although we used Z-FA-FMK, a broad-spectrum cysteine protease inhibitor, it is important to note that the observed effects cannot be solely attributed to cathepsin B inhibition. The results likely reflect the inhibition of multiple cysteine proteases, with cathepsin B being a key player among them. To further validate cathepsin B's specific role, we also utilized AAV-mediated overexpression of cathepsin B and its inhibitor cystatin

**Fig. 5 | AMs initiate matrix transfer by activating mesothelium through IL-18.**
**a** Representative immunolabeling histology images of IL-18 receptor expression in murine and human IPF lungs. $n$ = five biological replicates (C57BL/6J WT mice) and 3 independent experiments. Scale bars: 50 μm (murine); 20 μm (human).
**b** Representative immunolabeling histology image of IL-18 expression in lungs 1, 7, 10 and 14 days p.b.i. or 15 days post-herpes virus installation. $n$ = 3 biological replicates (C57BL/6J WT mice) and 3 independent experiments. Scale bars: 100 μm(10x); 50 μm(20x). Data represented are mean ± SD. A two-sided independent T-test was used for the comparison of two groups (p.b.i.: Day 1:$p$ = 0.000138, Day 7:$p$ = 0.000663, Day 14:$p$ = 0.0004998)(post-herpes virus: Day 15: $p$ = 8.34e-05, Day 45: $p$ = 7.34e-06)(***$P$ < 0.001). **c** Representative immunolabeling histology image showing contrast of IL-18 expression in lung interstitium compared to surface 14 days p.b.i. $n$ = five biological replicates (C57BL/6J WT mice) and 3 independent experiments. bars: 50 μm. **d** Representative immunolabeling histology images against IL-18 in combination with CD45 and CD11c in lungs 14 days p.b.i. or human IPF lung tissue. $n$ = 3 biological replicates (C57BL/6J WT mice) and 3 independent experiments. Data represented are mean ± SD. A two-sided independent T-test was used for the comparison of two groups (***$P$ < 0.001). Scale bars: 50 μm. **e** Representative histology images of mouse lung tissue with NHS-FITC$^+$ marked pleural site cultivated ex vivo with 10 ng/ml recombinant IL-18 or in trans wells with macrophages and 100 ng/ml IL-18 Receptor-blocking antibody for 48 h. Data represented are mean ± SD. $n$ = five biological replicates and 6 independent experiments. One-way ANOVA was used for the multiple comparison (***$P$ < 0.001). Scale bars: 25 μm. **f** Flowchart of ex vivo mouse lungs coculture with IL-18+ macrophages using transwell. Created in BioRender. Han, W. (2024) https://BioRender.com/y450579.

---

A, which supports our findings. Nevertheless, the use of more specific inhibitors or genetic approaches in future studies could help delineate the individual roles of these proteases. Lastly, while we use the term PAJs to facilitate discussion, we acknowledge that specific markers and precise definitions for PAJs remain provisional, requiring further research to clarify their role and contributions in fibrosis progression. Despite these limitations, our findings with chemical and viral injury models, reveal PAJs as a key entry point into fibrosis, and provide a promising direction to combat fibrotic diseases.

## Methods
### Patient derived tissue
Human tissue has been obtained from the CPC-M bioArchive at the Comprehensive Pneumology Center (CPC Munich, Germany). The study was approved by the local ethics committee of the Ludwig-Maximilians University of Munich, Germany (Ethic vote #333-10). Written informed consent was obtained for all study participants.

### Mouse housing and husbandry
C57BL/6J mice were purchased from Charles River and bred and maintained in the Helmholtz Animal Facility in accordance with EU directive 2010/63. IFN-γ-R$^{-/-}$ mice, on C57BL/6 background, were originally obtained from the Jackson Laboratory (Bar Harbor, ME, USA) and subsequently bred and propagated under SPF conditions at the Helmholtz Zentrum München. Animals were housed in individual ventilated cages and animal housing rooms were maintained at constant temperature and humidity with a 12-h light cycle. Animals were supplied with water and chow ad libitum. All animal experiments were reviewed and approved by the Government of Upper Bavaria and registered under the project number ROB-55.2-2532.Vet_02-19-101 or ROB-55.2-2532.Vet_02-18-97 and conducted under strict governmental and international guidelines. This study is compliant with all relevant ethical regulations regarding animal research.

### In vivo matrix fate tracing
We generated a labeling solution by mixing 5 μl Succinimidyl ester (N-Hydroxysuccinimide (NHS)-ester; Thermo Fisher) (25 mg/ml) in DMSO with 5 μl of 100 mM pH 9.0 sodium bicarbonate buffer plus 40 μl PBS to a total volume of 50 μl. Labeling solution was applied intrapleurally under isoflurane anesthesia with a 30G cannula. In the case of abdominal tracing, 100 μl of the labeling solution was administered intra-peritoneally. NcAAs (Azidohomoalanine (AHA), Homo-propargylglycine (HPG); Thermo Fisher) were injected intra-peritoneally (IP) with 0.025 mg/g per day.

### Bleomycin induced pneumonia model
Bleomycin was administered 24 h after labeling the pleural matrix. Oropharyngeal administration of bleomycin was performed in conjunction with anesthesia in C57BL/6J mice of both sexes (6–8 weeks age). After verifying absence of toe-pinch reflex in anesthetized mice, mice were placed on a restraining table and kept in an upright position. The tongue was carefully fixed and held to the side with tweezers and the nose of the animal was covered with tweezers. By keeping the nose closed, the mouse is forced to breathe through the mouth. With the help of a pipette, bleomycin was dissolved in a dosage of 2 units/kg KGW in 80 μl PBS carefully into the throat. As soon as the animal had inhaled the solution, it was transferred on to a hot plate (duration ~30 to 60 s) and subsequently housed for indicated days.

### Pharmacologic regime
Cathepsin Inhibitor B (0.15 mg/kg) were injected intra-peritoneally, 1 h before bleomycin installation and every other day, in a volume of 100 μl in physiological saline solution.

### Herpes induced pneumonia model
Mice were housed in individually ventilated cages during the MHV−68 infection period. Mice were infected intranasally with 5 × 10*4 plaque-forming units of MHV-68 diluted in PBS in a total volume of 30 μl. Prior to intra-nasal infection, mice were anesthetized with medetomidine–midazolam–fentanyl and labeling solution was applied to label organ surfaces as described above. At the predetermined time points, mice were sacrificed by cervical dislocation and tissues were processed for subsequent experiments.

### Recombinant TGF model
100 ng of recombinant TGF-β was applied intrapleural under isoflurane anesthesia with a 30G cannula.

### Plasmid construction
To construct the plasmids for AAV production, cDNA was generated from mRNA extracted from C57BL/6 mice tissue utilizing SuperScript IV Reverse Transcriptase (Life Technologies). Produced cDNA was utilized as template for PCRs amplifying coding sequences of murine TGFβ (TGFb), cathepsin B (CTSB) and cystatin A (CSTA) using KOD Hot Start DNA Polymerase (Merck Millipore). Analog, the dominant negative mutant of murine TGFβRII (DN-TGFbRII) according to the published human version was produced[51]. Flag-tagged murine collagen 1a2 (Col1a2-Flag) was created based on plasmid eGFP-proα2(I) (gifted by Sergey Leikin; Addgene plasmid # 119826; http://n2t.net/addgene:119826; RRID:Addgene_119826) replacing eGFP with the Flag-tag coding sequence. PCR products were cloned into pAAV-Cp-SV40pA containing CMV immediate early promoter and SV40polyA sequence flanked by AAV2 derived inverted terminal repeats (ITRs) utilizing In-Fusion HD Cloning Plus (Takara Bio Inc.). Final plasmids were verified by sequencing. For generation of capsid-modified AAV8RGD the plasmid pAAV2/8, a gift from James M. Wilson (Addgene plasmid # 112864; http://n2t.net/addgene:112864; RRID:Addgene_112864), was modified by incorporation of peptide TGCDCRGDCFCG between amino acid 584 and 585 of VP1. Final plasmid pAAV2/8RGD was

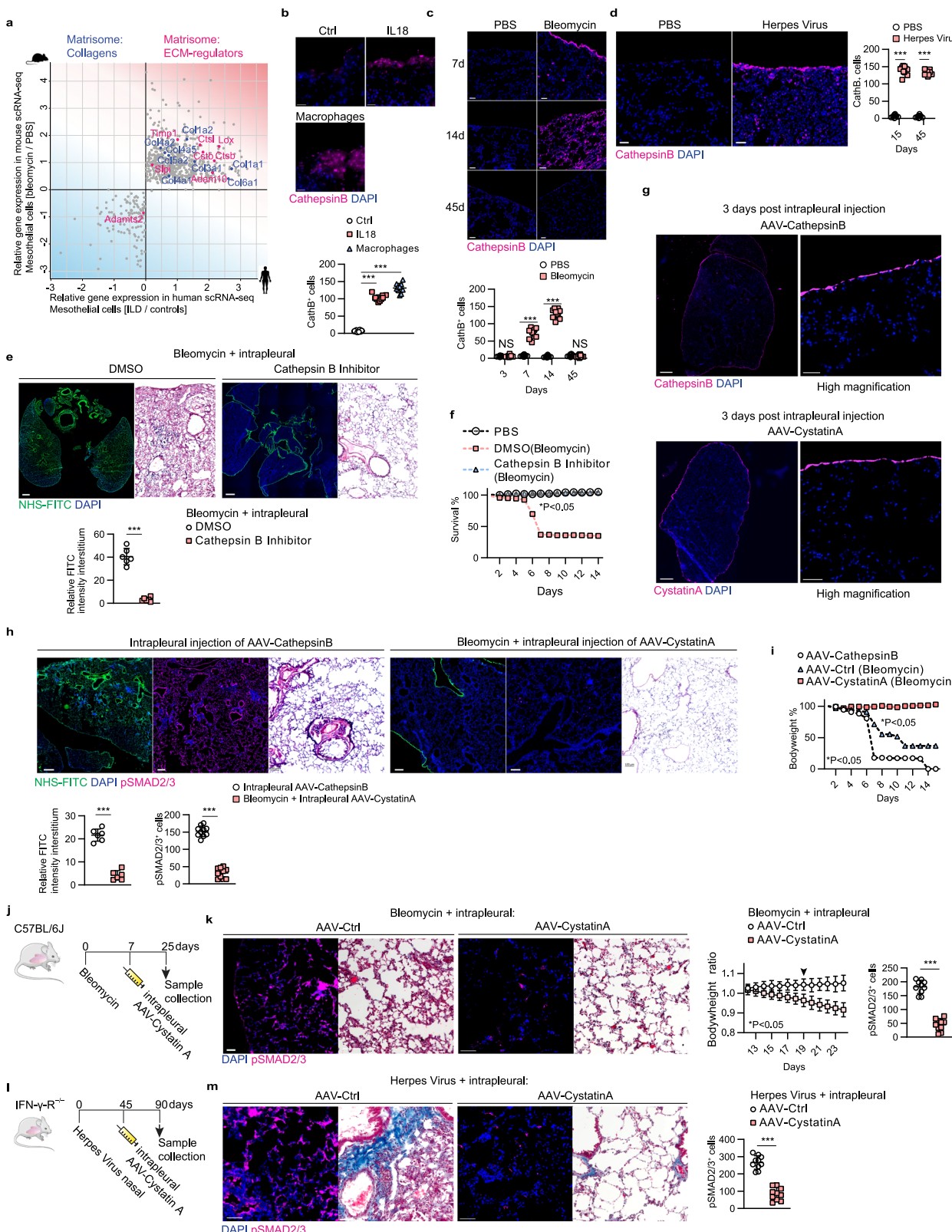

verified by sequencing. Sequences were uploaded into our repository. Details of the plasmid construction procedure can be obtained upon request.

## AAV production

Production and purification, of AAV-preparations for AAV8RGD-TGFb, AAV8RGD-DN-TGFbRII, AAV8RGD-CTSB, AAV8RGD-CSTA and AAV8RGD-Col1a2-Flag, was performed according to the AAVpro®

Purification Kit Maxi (Takara Bio Inc.) protocol. In brief, 5x T225-flasks were triple-transfected with (i) the pHelper plasmid from the AAVpro® Helper Free System (AAV6) kit (Takara Bio Inc.), (ii) the plasmid pAAV2/8RGD containing coding sequences of the AAV2-derived rep proteins and the modified AAV8 capsid proteins and (iii) the pAAV-Cp-SV40pA derivate containing the AAV-genome with the respective transgene. 96 h post transfection cells were harvested and AAV vector particles were released by breaking up the cells with

**Fig. 6 | Mesothelial Cathepsin B drives pleural matrix transfer and pulmonary fibrosis. a** scRNA-Seq data from bleomycin-installed mice and ILD patients. **b** Representative immunolabeling of Cathepsin histology images of ex vivo mouse lungs treated with IL-18 or transwell coculture with IL-18^high Macrophages. $n = 10$ biological replicates (C57BL/6J WT mice) and 5 independent experiments. Scale bars: 20 µm. **c, d** Representative immunolabeling histology images of Cathepsin B in mouse lungs 7, 14 and 45 days p.b.i. or 15 days post-herpes virus. $n = 10$ biological replicates (C57BL/6J WT mice or IFN-γ-R^-/^- mice) and 5 independent experiments. p.b.i.: Day 7: $p = 5.71e-10$, Day14: $p = 8.59e-11$; post-herpes virus: Day15: $p = 7.32e-12$, Day 45: $p = 1.59e-16$. Scale bars: 50 µm. **e** Representative histology images of mouse lungs 14 days p.b.i. Mice were intrapleurally injected with NHS-FITC labelling mix. The next day bleomycin was installed and cathepsin B inhibitor were applied every other day in 10 µm concentration, DMSO acted as control. Log-rank test was used for statistical comparison. $n = 6$ biological replicates (C57BL/6J WT mice) and 3 independent experiments. Scale bars: 1000 µm (Fluorescence); 100 µm (Histology). **f** Survival of mice in (**e**). Log-rank test was used for statistical comparison. **g** Representative immunolabeling histology images of Cathepsin B and its inhibitor Cystatin A in mouse lungs 3 days post-intrapleural AAV injection. $n = 6$ biological (C57BL/6J WT mice) and 6 independent experiments. Scale bars: 500 µm; high magnification: 20 µm. **h** Representative histology images of mouse lungs 14 days p.b.i. Mice were intrapleurally injected with NHS-FITC labelling mix. AAV particles encoding for Cathepsin B inhibitor Cystatin A were also applied intrapleurally; then, five days later bleomycin was installed. For Cathepsin B coding AAVs no bleomycin

was applied. $n$= six biological replicates (C57BL/6J WT mice) and 3 independent experiments. Masson trichrome was used to visualize structural changes, pSMAD2/3 staining visualized fibroblast activation. FITC: $p = 1.69e-07$, pSAMD: $p = 6.84e-13$. Scale bars: Fluorescence 1000 µm (Overview); 100 µm (pSMAD2/3 staining); Masson Trichrome 100 µm. **i** Bodyweight and survival of mice in (**h**). Log-rank test was used for statistical comparison. **j** Schematic representation of AAV-Cystatin mediated intervention in bleomycin lung fibrosis model experiment. Mice were treated with bleomycin and seven days later, mice were intrapleurally injected with AAV-Cystatin A. **k** Representative histology- and pSMAD2/3 immuno- staining images 24 days and bodyweights of animals after bleomycin treatment. $n = 6$ biological replicates (C57BL/6J WT mice) and 3 independent experiments. Scale bars: immunostainings: 100 µm; Masson trichrome 100 µm. Log-rank test was used for statistical comparison of bodyweights. pSAMD: $p = 9.46e-11$. **l** Schematic representation of AAV-Cystatin mediated intervention in herpes virus lung fibrosis model. Mice were treated with herpes virus and 45 days later, mice were intrapleurally injected with AAV-Cystatin A. **m** Representative histology- and pSMAD2/3 immuno- staining images 90 days after Herpes virus treatment. $n = 6$ biological replicates (IFN-γ-R^-/^- mice) and 3 independent experiments. Scale bars: immunostainings: 100 µm; Masson trichrome 50 µm. pSAMD: $p = 2.54e-09$. All data represented in Fig. 6 are mean ± SD. One-way ANOVA was used for the multiple comparison. Single comparison was performed by two-sided independent T-test. (***$P < 0.001$; NS= not significant).

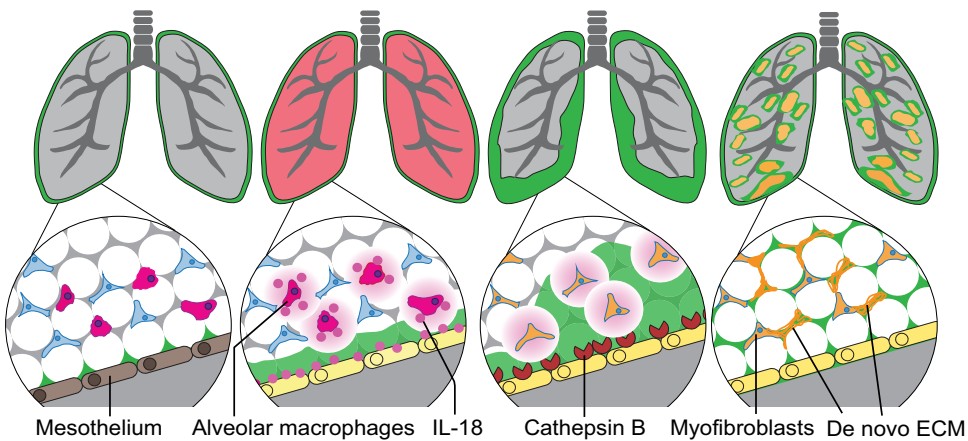

Mesothelium    Alveolar macrophages    IL-18        Cathepsin B    Myofibroblasts    De novo ECM

**Fig. 7 | Revised model of lung fibrosis development.** Schematic representation of lung fibrosis development. **a** A monolayer of mesothelium (shown in pink) encapsulates a thin layer of matrix reservoir (shown in green) in healthy lungs. **b** Lung injury activates alveolar macrophages to secrete IL-18. IL-18 triggers surface

mesothelium through IL-18 receptor. **c** Surface mesothelium produces Cathepsin B that liberates pleural matrix pools, triggering matrix transfer inwards. **d** Invaded matrix changes interstitial environment, which in turn activates fibroblasts and forming of pulmonary fibrosis.

3x freeze-thaw cycles. Genomic DNA was digested with Cryonase cold-active nuclease and AAV vector particles were separated from cell debris by filtration (0.45 µm filter). Finally, AAV particles were separated from low molecular weight contaminants utilizing 100 kDa size exclusion columns, and concentrated. Titers of final AAV preparations were determined via qPCR utilizing the AAVpro Titration Kit (qPCR) V2 (Takara Bio Inc.).

### AAV application in mice
For application of AAVs in mice, viral vector preparations were diluted with PBS (1x) to a final concentration of $6 \times 10^8$ viral particles/µl. 50 µl of respective vector dilutions were used for intrapleural injections (total dose of $3 \times 10^{10}$ viral particles).

### Ex vivo culture of lung biopsies
C57BL/6J male mice (6–8 weeks age) were used to study the movement of lung matrix. After organ withdrawal, 4 mm biopsy punches of murine lungs were generated. To obtain ectopic labeling of matrix, we generated a labeling solution by mixing NHS-ester 1:1 with 100 mM pH

9.0 sodium bicarbonate buffer. Sterile Whatman filter paper (Sigma Aldrich) biopsy punches were soaked in NHS-labeling solution, and locally placed on the lung biopsy surface. After 1 min, the labeling punch was removed. Mouse lung biopsies were cultured in the RPMI 1640 medium (10% FBS with 1% Pen/Strep and 0.1% AmB). Mouse lung biopsies with immune cells were then cultured in ex vivo conditions, provided with 5% $CO_2$ at 37 °C. After 48 h, mouse lung biopsies were fixed with the 4% PFA and incubated overnight at 4 °C followed by a PBS wash. Peritumor human lung tissues ($n = 5$) and IPF human lung tissues ($n = 5$) were obtained from hospital, and then labeled by mixing NHS-ester 1:1 with 100 mM pH 9.0 sodium bicarbonate buffer, and cultivated for 24 h as described above.

In the transwell assay, mouse lung biopsies were put in the 0.4 µm pore polyester membrane transwell insert (Corning, 3470) and the media in the bottom chamber was supplemented with chemical compounds or macrophages. Both media in the transwell insert and bottom chamber were supplied with RPMI 1640 medium (10% FBS with 1% Pen/Strep and 0.1% AmB). The chemical compounds include recombinant IL18 (10 ng/ml) and IL18 blocking antibody (100 ng/ml).

## Tissue preparation histology

Upon organ excision, they were fixed overnight at 4 °C in 2% formaldehyde. The next day, fixed tissues were washed three times in Dulbecco's phosphate buffered saline (DPBS, GIBCO, #14190-094), and depending on the purpose, either embedded, frozen in optimal cutting temperature (OCT) compound (Sakura, #4583) and stored at −20 °C, or stored at 4 °C in PBS containing 0.2% gelatin (Sigma Aldrich, #G1393), 0.5% Triton X-100 (Sigma Aldrich, #X100) and 0.01% Thimerosal (Sigma Aldrich, #T8784) (PBS-GT). Fixed tissues were embedded in OCT and cut with a Microm HM 525 (Thermo Scientific). In brief, sections were fixed in ice-cold acetone for 5 min at −20 °C, and then washed with PBS. Sections were then blocked for non-specific binding with 10% serum in PBS for 60 min at room temperature, and then incubated with primary antibody in blocking solution O/N at 4 °C. The next day, following washing, sections were incubated in PBS with fluorescent secondary antibody, for 120 min at RT. Finally, sections were washed and incubated with Hoechst 33342 nucleic acid stain (Invitrogen, #H1399), washed in ddH$_2$O, mounted with Fluoromount-G® (Southern Biotech, #0100-01), and stored at 4 °C in the dark. Visualization of ncAAs was performed using Alkyne-Alexa Fluor 647 (Thermo Fisher) and Click-iT Cell Reaction Buffer Kit (Thermo Fisher).

## 3D multiphoton imaging

For multi-photon imaging, samples were embedded in a 4% NuSieve GTG agarose solution (Lonza, #50080). Imaging was performed using a 25x water-dipping objective (HC IRAPO L 25x/1.00 W) coupled to a tunable pulsed laser (Spectra Physics, Insight DS+). Multiphoton excited images were recorded with external, non-descanned hybrid photo detectors (HyDs). Then, band-pass (BP) filters were used for detection: HC 405/150 BP for Second Harmonic Generation (SHG) and an ET 525/50 BP for green channel. Tiles were merged using Leica Application suite X (v3.3.0, Leica) with smooth overlap blending. Finally, data were visualized with Imaris software (v9.1.3, Bitplane).

## 3D light sheet imaging

Whole-mount samples were stained and cleared with a modified 3DISCO protocol[52]. Samples were dehydrated in an ascending tetrahydrofuran (Sigma Aldrich, #186562) series (50%, 70%, 3 × 100%; 60 min each), and subsequently cleared in dichloromethane (Sigma Aldrich, #270997) for 30 min and eventually immersed in benzyl ether (Sigma Aldrich, #108014). Cleared samples were imaged, whilst submerged in benzyl-ether, with a light-sheet fluorescence microscope (LaVision BioTec). Whilst submerged in benzyl-ether, specimens were illuminated on two sides by a planar light-sheet using a white-light laser (SuperK Extreme EXW-9; NKT Photonics). Optical sections were recorded by moving the specimen chamber vertically at 5-mm steps through the laser light-sheet. Three-dimensional reconstructions were obtained using Imaris imaging software (v9.1.3, Bitplane).

## Histology and murine ex vivo imaging

Histological sections were imaged under a M205 FCA Stereomicroscope (Leica) and ZEISS AxioImager Z2m (Carl Zeiss). Murine biopsy punches were imaged under a M205 FCA Stereomicroscope (Leica). Data was processed with Imaris 9.1.3 (Bitplane) and ImageJ (1.52i). Contrast and brightness were adjusted for better visibility.

## Image quantification

Matrix invasion was calculated using histological sections, quantifying FITC signal per area via ImageJ (1.52i). Quantification of immunolabeling was performed in randomly distributed ROIs/FOVs. Multiphoton images were analyzed using Fiji plugins Fraklac V. 2.5 and LocalThickness_V-4.0.2[53]. Overlay channels were generated using Imaris 9.1.3 (Bitplane).

## Monocyte isolation and differentiation

Primary Monocytes were isolated from fresh blood from mouse femur and tibia using a monocyte isolation kit (Miltenyi Biotec), followed by red blood cell lysis and immunomagnetic depletion of labeled non-target cells, such as T cells, B cells, NK cells, dendritic cells, erythroid cells, and granulocytes. Subsequently, monocyte subsets expressing CD14+, CD16- were purified and collected in flow-through buffer. RPMI 1640 (GIBCO) supplemented with 10% (v/v) FBS (Sigma), 20 ng/ml GM-CSF (Sigma), 1% (v/v) Pen-strep (Sigma), and 1% (v/v) sodium pyruvate (GIBCO) was used to culture the cells. The suspended monocytes were placed in a 37 °C incubator with 5% CO$_2$ and differentiated into adherent macrophages after 7 days. Then the macrophages were collected for purification of IL-18+ macrophages. The protocol was performed by using Magnetic-Activated Cell Sorting (Miltenyi biotech, 130-115-674) according to the manufacturer's instructions (Miltenyi biotech, 130-090-485 and 130-042-401).

## Protein biochemistry

To separate the pleura from deep interstitial tissues, firstly, the tissue sample was carefully isolated to expose the pleural surface. Subsequently, gentle traction and blunt dissection was applied to gradually separate the pleura from the underlying interstitial tissues, taking care to avoid inadvertent damage to either structure. Tissues were snap-frozen and ground-up using a tissue lyser (Quiagen). Pulverised tissues were resuspended in lysis buffer (20 mM Tris-HCl pH 7.5, 1% Triton X-100, 2% SDS, 100 mM NaCl, 1 mM sodium orthovanadate, 9.5 mM sodium fluoride, 10 mM sodium pyruvate, 10 mM beta-glycerophosphate), and supplemented with protease inhibitors (complete protease inhibitor cocktail, Pierce) and kept 10 min on ice. Samples were sonicated and spun down for 5 min at 10,000 × g. Supernatants were stored at −80 °C. Protein concentration was determined via BCA-Assay according to manufactures protocol (Pierce).

For further MS analysis, protein pulldown was performed as follows, which is only to extract proteins labeled with NHS-EZ-LINK from both pleura and interstitum. Lysates were diluted with a pulldown buffer (20 mM Tris-HCl pH 7.5, 1% Triton X-100, 100 mM NaCl, supplemented with protease and phosphatase inhibitors) and incubated overnight with Dynabeads M-270 Streptavidin at 4 °C on a rotator according to the manufacturer's instructions. The next day, the samples were each washed twice with Wash Buffer 1 (20 mM Tris-HCl pH 7.5, 1% Triton X-100, 2% SDS, 100 mM NaCl, supplemented with protease and phosphatase inhibitors) and then with Wash Buffer 2 (20 mM Tris-HCl pH 7.5, 0.5% Triton X-100, 100 mM NaCl, supplemented with protease and phosphatase inhibitors) and finally washed twice with Wash Buffer 3 (20 mM Tris-HCl pH 7.5 and 100 mM NaCl). Beads were then resuspended in Elution Buffer (20 mM Tris-HCl pH 7.5, 100 mM NaCl and 50 mM DTT) and incubated for 30 min at 37 °C. Finally, the samples were boiled for 5 min at 98 °C and the supernatants were stored at −80 °C.

## Mass spectrometry

Tissue lysis was performed as described above. Samples were digested using a modified FASP procedure[54]. After reduction and alkylation using DTT and IAA, the proteins were centrifuged on Microcon® centrifugal filters (Sartorius Vivacon 500 30 kDa), washed thrice with 8 M urea in 0.1 M Tris/HCl pH 8.5 and twice with 50 mM ammonium bicarbonate. The proteins on filters were digested for 2 h at room temperature using 0.5 µg Lys-C (Wako Chemicals) and for 16 h at 37 °C with 1 µg trypsin (Promega). Peptides were collected by centrifugation (10 min at 14,000 × g), acidified with 0.5% TFA and stored at −20 °C until measurements. The digested peptides were loaded automatically on an HPLC system (Thermo Fisher Scientific) equipped with a nano trap column (100 µm ID ×2 cm, Acclaim PepMAP 100 C18, 5 µm, 100 Å/size, LC Packings, Thermo Fisher Scientific) in 95% buffer A (2% ACN, 0.1% formic acid (FA) in HPLC-grade water) and 5% buffer B (98% ACN,

0.1% FA in HPLC-grade water) flowing at 30 μl/min. After 5 min, the peptides were eluted and separated on the analytical column (nanoEase MZ HSS T3 Column, 100 Å, 1.8 μm, 75 μm × 250 mm, Waters) for 105 min at 250 nl/min flow rate in a 3 to 40% non-linear acetonitrile gradient in 0.1% formic acid. The eluting peptides were analyzed online in a Q Exactive HF mass spectrometer (Thermo Fisher Scientific) coupled to the HPLC system with a nano spray ion source, operated in the data-dependent mode. MS spectra were recorded at a resolution of 60,000 and after each MS1 cycle, the 10 most abundant peptide ions were selected for fragmentation. Raw spectra from mouse samples were analyzed with Progenesis QI software (version 4.1, Nonlinear Dynamics, Waters) and searched against the SwissProt mouse database (16,872 sequences) with Mascot (Matrix Science, version 2.6.2) with the following search parameters: 10 ppm peptide mass tolerance and 0.02 Da fragment mass tolerance, two missed cleavages allowed; carbamidomethylation was set as fixed modification, camthiopropanoyl, methionine and proline oxidation were allowed as variable modifications. A Mascot-integrated decoy database search calculated an average false discovery of <5% when searches were performed with a mascot percolator score cut-off of 13 and a significance threshold p-value[55]. Peptide assignments were re-imported into the Progenesis QI software and the abundances of all unique peptides allocated to each protein were summed and normalized. Raw spectra from human samples were analyzed with Proteome Discoverer 2.4 software (Thermo Fisher Scientific; version 2.4.1.15) via a database search (Sequest HT search engine) against the SwissProt human database (20,237 sequences), considering full tryptic specificity, allowing for up to two missed tryptic cleavage sites, precursor mass tolerance 10 ppm, fragment mass tolerance 0.02 Da. Carbamidomethylation was set as fixed modification; camthiopropanoyl, methionine and proline oxidation were allowed as variable modifications. Percolator was used for validating peptide spectrum matches and peptides, accepting only the top-scoring hit for each spectrum, and satisfying the cutoff values for FDR of <1%, and posterior error probability of <0.05[55]. The final list of proteins, complied with the strict parsimony principle and, contains the summed and normalized abundances of all qualifying peptides.

Extracellular elements were identified through a database search against a matrix gene database[56]. Gene ontology analysis was performed using EnrichR webtool[57,58].

### scRNA-Seq analysis

Single cell sequencing data of whole mouse lung lysates from a bleomycin time course experiment was re-analyzed regarding the Mesothelial cells[35]. The dataset includes 28 samples and 29,297 cells, with 652 mesothelial cells. Differentially expressed genes across timepoints within the mesothelial cells were identified, using the R packages *splines* and *lmtest*[35]. Differentially expressed genes between PBS controls and bleomycin-induced lung fibrosis samples, were calculated using the *allMarkers* function of *scanpy*. These genes were compared with the differentially expressed genes between human ILD patients and controls within mesothelial cells, from an integrated human ILD lung cell atlas[59].

Ligand–receptor analyses were performed on the basis of the R package CellChat (v1.6.0)[60]. We computed the cell communication probability between each module and other cell populations using the CellChat function computeCommunProb. The overall scaled communication probability was then visualized, based on a circle plot, using a customized plot_communication function. To further understand which signals contribute most to the ligand–receptor (LR) interaction pathways for the mesothelial cells, we clustered the communication patterns of the cells populations using the function identifyCommunicationPatterns. We then computed the cell populations' incoming and outgoing signal using the function netAnalysis_signalingRole_scatter. We also visualized the IL-18 signaling pathway network using the function netAnalysis_signalingRole_network.

For Human interaction analysis we used the human lung atlas[26]. We extracted the alveolar macrophages and annotated them according to their expression of IL-18 (>10 was considered IL-18 high). We then analyzed the Enriched pathways of the IL-18 high cluster using gseapy python package (https://doi.org/10.1093/bioinformatics/btac757). We also performed cell-cell analysis according to murine and generated a heatmap for network signaling score for all the significant pathways using the function netAnalysis_signalingRole_network.

### Quantification and statistical analysis

All experiments were performed with at least three biological repeats. Results are expressed as mean ± s.d. or mean ± s.e.m., statistical tests are indicated in figure legends. A *P* value of <0.05 was considered statistically significant. Analyses were conducted using GraphPad 8.0 software.

### Reporting summary

Further information on research design is available in the Nature Portfolio Reporting Summary linked to this article.

### Data availability

Source data are provided with this paper. All datasets generated in this study have been deposited and are available as follows: https://zenodo.org/records/3865110?preview=1&token=eyJhbGciOiJIUzUxMiJ9.eyJpZCI6ImQ1NjllNWE0LTI1MjUtNGJlMi1iNWI3LWFhZDdjNjVlYzM3MCIsImRhdGEiOnt9LCJyYW5kb20iOiJkMGQ3M2ZkOGEyN2M5YmFhNzEyNzZjMTFiODNmNDRiOCJ9.IsaPo7WJEw5Yz1pZ11bDHThgqDZdyxDPTGfAp02bPDx9n5_k7s; https://zenodo.org/records/4294005?preview=1&token=eyJhbGciOiJIUzUxMiJ9.eyJpZCI6IjJhYTQwODMyLWEzMzQtNDg2YS1iOGFkLTExMTRkNTUwZDg3MyIsImRhdGEiOnt9LCJyYW5kb20iOiJyZGUwNjkzYWE3MDA1YjFkYWM3NjE3NzM0MmMwOWY4YiJ9.4Z4OBT-H40DzMFlLugprYwsaUzq2fK8BGMs6vPZ3zjCdHXTTt7; https://zenodo.org/records/14357585?preview=1&token=eyJhbGciOiJIUzUxMiJ9.eyJpZCI6IjdhNWUwNWMwLTExMzQtNGU4NC1iMjdkLTE0ZjA0ODk3NDVhOCIsImRhdGEiOnt9LCJyYW5kb20iOiJkZWE5MWYzMzllMmM5Y2UzYzU4ZTA4MGQ1YjUxZTUxNyJ9.gBGN8wJWbSrAvvrfGg1fxhhhlxgdSf1lxjCmWkb5_JGVwgrjo; Additional information is available from the corresponding author on request. Source data are provided with this paper.

### Code availability

Computer codes are available from the corresponding author on request.

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

## Acknowledgements

We thank S. Dietzel and the Core Facility Bioimaging at the Biomedical Centre of the Ludwig-Maximilians Universität München for access and support with the multi-photon system. We gratefully acknowledge the provision of human biomaterial and clinical data from the CPC-M bioArchive and its partners at the AsklepiosBiobank Gauting, the Klinikum der Universität München and the Ludwig-Maximilians-Universität München. We thank the animal caretakers of Unit34 and SMAP for their constant support and care of our mice. Y.R. was supported by the Human Frontier Science Program Career Development Award (CDA00017/2016), the German Research Foundation (RI 2787/1-1 AOBJ: 628819), the Fritz–Thyssen–Stiftung (2016-01277), the Else-Kröner-Fresenius-Stiftung (2016_A21) and the European Research Council Consolidator Grant (ERC-CoG 819933). A.F. was supported by a PFP - Helmholtz Postdoctoral Fellowship.

## Author contributions

Y.R. and A.F. supervised the research narrative. A.F., J.W., and Y.R. designed bleomycin and viral experiments. A.F., J.W., M.M.H., and S.C. performed bleomycin animal installations. A.F. and M.M.H. designed and performed AAV based experiments. M.M.H. designed, cloned and purified AAV-based vectors. A.F., M.M.H., S.C., and H.A. performed herpes virus experiments. A.F. designed and performed protein biochemistry experiments. S.C., T.G.G., and C.D. performed multiphoton microscopy. S.C. performed light-sheet microscopy and generated all supplementary videos. A.F., W.H., and S.H. performed, imaged and analyzed histology and immunofluorescence staining. Y.S., B.D., A.S., and S.D. assisted with the histology staining and revision process. W.H. and T.G.G. cultivated human tissue experiments. W.H., Y.L., and R.D. performed murine ex vivo assays. Y.L. and R.D. established and performed macrophage preparations. R.H., M.G.S., M.L. and A.H. provided human tissue. J.W. and A.F. provided veterinary advice and prepared animal experiment protocols. S.M.H., W.H., A.F., S.K. performed and analyzed mass spectrometry experiments. W.H. and S.H. assisted with the histology sections. S.K., A.K., C.H.M., and H.B.S. performed scRNAseq analysis. H.G.M. provided support and assisted in clinical interpretation of the animal data. A.F., W.H., and S.H. generated figures. A.F. and D.C.G. generated artistic illustrations. A.F., W.H., S.H., and Y.R. wrote the manuscript, and W.H., S.H., and Y.R., revised the manuscript.

## Funding

## Competing interests

The authors declare the following competing interests: A.F., M.M.H., S.K., and Y.R. have filed patent application EP21206 688.0 covering the use of these methods to study extracellular matrix movement in organ fibrosis. The remaining authors declare no competing interests.

## Additional information

[1]Institute for Diabetes and Obesity (IDO), Helmholtz Diabetes Center (HDC), Helmholtz Zentrum München, Neuherberg, Germany. [2]Institute of Regenerative Biology and Medicine(IRBM), Helmholtz Zentrum München, Munich, Germany. [3]Member of the German Center of Lung Research (DZL), Munich, Germany. [4]Faculty of Medicine, Ludwig-Maximilians-University Munich, Munich, Germany. [5]Zhangzhou Health Vocational College, Zhangzhou, China. [6]Helmholtz Munich, Research Unit for Precision Regenerative Medicine (PRM), Member of the German Center for Lung Research (DZL), Munich, Germany. [7]Institute for Stroke and Dementia Research (ISD), LMU University Hospital, LMU Munich, Munich, Germany. [8]Asklepios Fachkliniken in Munich-Gauting, Munich, Germany. [9]University Department of Visceral and Thoracic Surgery Salzburg, Paracelsus Medical University, Salzburg, Austria. [10]Helmholtz Zentrum München, Institute of Lung Biology & Disease, Group Mechanism of Neonatal Chronic Lung Disease, Member of the German Center of Lung Research (DZL), Munich, Germany. [11]Comprehensive Pneumology Center with the CPC-M bioArchive and Institute of Lung Health and Immunity, Helmholtz-Zentrum München, Member of the German Center of Lung Research (DZL), Munich, Germany. [12]Institute of Asthma and Allergy Prevention, Helmholtz Zentrum München, German Research Center for Environmental Health, Neuherberg, Germany. [13]Walther-Straub-Institute of Pharmacology and Toxicology, Ludwig-Maximilians-University Munich,

Munich, Germany. [14]Department of Plastic and Hand Surgery, Technical University of Munich, School of Medicine and Health, Klinikum rechts der Isar, Munich, Germany. [15]Institute of Experimental Pneumology, LMU University Hospital, Ludwig-Maximilians University, Munich, Germany. [16]Metabolomics and Proteomics Core, Helmholtz Zentrum München, Munich, Germany. [17]Institute of Regenerative Biology and Medicine, Chinese Institutes for Medical Research, Beijing, China. [18]Capital Medical University, Beijing, China. [19]These authors contributed equally: Adrian Fischer, Wei Han, Shaoping Hu, Martin Mück-Häusl, Juliane Wannemacher. ✉e-mail: wei.han@helmholtz-munich.de; yuval.rinkevich@cimrbj.ac.cn

