## [Transparent Peer Review file · Nature Communications]

Targeting pleuro-alveolar junctions reverses lung fibrosis in mice

Corresponding Author: Dr Wei Han

Version 0:

Reviewer comments:

Reviewer #1

(Remarks to the Author)

This research is quite interesting and the findings are novel. These investigators explored the role of macrophages and found that these cells transfer connective tissue from pleuro-alveolar (extracellular matrix underneath the mesothelial lining) junctions to lung parenchyma. They found that pleuro-alveolar junction macromolecular transfer of connective tissue was present in human lung biopsies and chronic models of lung fibrosis, and was reduced in acute fibrosis models. Moreover, this process could be inhibited by Cystatin (cysteine protease inhibitor). However, several concerns need to be addressed by the investigators to improve the rigor and causality of the observations.

1. Imaging is not yet convincing. the investigators should provide more specifics (imaging parameters including exposures, antibodies used), the FITC channel has a lot of autofluorescence - how is distinguished from collagen/elastin, metadata should be included, and the human lung findings need to be better defined histologically.
2. Specificity: Is this inward movement specific to ECM or would any abnormal protein or particle be similarly phagocytosed by macrophages and moved centrally?
3. Causality: The key observation is that macrophages are moving ECM from the pleura to the more central portions of the lung. While the investigators present convincing findings to support this finding, it's unclear whether the movement of ECM from the pleura to the central portions of the lung are responsible for the development of lung fibrosis. Additional experiments are needed to prove causality.

Reviewer #2

(Remarks to the Author)

In this manuscript, Fischer et al. follow their own similar observations in skin and liver fibrosis on "mobile ECM" and suggest that mesothelial made ECM translocate to the interstitium (with unknown mechanisms) promoting pulmonary fibrosis. The authors suggest that the process is facilitated by macrophage derived IL-18 promoting cathepsin B expression from mesothelial cells. However, there are several conceptual as well as methodological issues that need to be addressed.

Major

Pleuro-alveolar junctions should be better defined (including some molecule markers, as well as hypothesized functions), as I could not find any relevant publications in the literature; the cited references are not relevant.

The intriguing concept of "ECM movement" should be better introduced and discussed. The previous publications from the same group, which decrease novelty, should be further discussed considering the findings here. Some questions should be entertained: how is it possible for such a highly complex and linked macromolecular structure to move around? By which mechanisms? Passive or energetic? In their previous publications suggested that fibroblasts can drag ECM cargo around. How?

Although mesothelial cells and their ECM production has been reported to play a role in pulmonary fibrosis (mostly in humans), that should be cited btw, do mesothelial cells secrete ECM in the pleural cavity (besides the lung underneath) where NHS is administered? What for?

NHS-FITC leakage upon the well-known bleomycin-induced increased endothelial/epithelial/mesothelial permeability is not shown or examined and could account for additional labelling post BLM. Intrapleural administration of Evans blue followed by bleomycin should be performed to examine leakage; Ext Fig 1b does not prove much.

In all imaging figures/panels, that the paper relies on, proof of claimed colocalizations (orthogonal and k-curve analyses) should be provided. The specificity of all antibodies should be provided (isotype and dilution controls). Higher/lower magnifications should also be provided; conclusions should not be based on one shown high magnification image; more optical fields, intensity quantifications and statistics are needed.

The choice of the herpes virus model is not explained sufficiently. Adenoviral delivery of TGF would be a much better choice, eliminating inflammatory contributions (and endo/meso-thelial leakage) to "ECM transfer". Moreover, bleomycin effects are not fully analyzed and thus the drawn conclusions are questionable. Although IT/OA administration of bleomycin is the most frequent route of administration, the authors should administer BLM (also) intravenously (IV), since with this method fibrosis develops subpleurally (as in humans, and as opposed to perivascular development upon IT administration), and therefore much better for the purposes of this paper.

Results from the re-analysis of publicly available scRNAseq datasets should be mostly supplementary and not main figures (that should contain novel data). Moreover, the results of the re-analyses can only be used for hypothesis generation and further validation experiments are necessary to substantiate any claims.

Does exposure of mesothelial cells (some established line or primary cells) to IL-18 promote Cathepsin B expression?

Z-FA-FMK is an inhibitor of not only Cathepsin B, but also cathepsins L, and S, cruzain, and papain. Most importantly, Z-FA-FMK also selectively inhibits effector caspases 2, 3, 6, and 7.

Terminology (Pleuro-alveolar junctions, mobile ECM, fate mapping, fibroblast licensing, fluidity factor) should be used with caution.

Minor

The definition of macrophage subpopulations is too general. Alveolar vs interstitial, and resident vs inflammatory are very important distinctions, especially in the context of the paper. The statement "IL-18 expressing macrophages are the major immune cell population" is not sufficiently supported by the literature.

All cited references should be reevaluated: some are irrelevant, some are outdated, while some cited reviews are low impact publications; much more authoritative reviews exist.

Second harmonic signal microscopy should be introduced. The SHG co-localization with the FITC signal should be proved. What does the suggested co-localization prove and why?

Fibroblast identification in immunocytochemistry is not sufficiently convincing. Which PDGFR was used? PRFGRA and PDGRB mark distinct fibroblast subpopulations with opposing functions. Positive/negative controls, lower magnifications, proof of colocalization, intensity quantification and statistics are needed. The same applies to all imaging figures in the manuscript.

There is discrepancy regarding BLM administration: the text mentions intratracheal injection (IT) and the methods oropharyngeal (OA).

If NHS-FITC tagging post BLM is "linked to myofibroblast activation" how do the authors explain the lighting up of trachea and bronchi? Larger area images are also needed.

There is a discrepancy concerning Extended figure 1d-e between results' text and figure legend. Liver or lungs? Ext Fig. 1f should be better explained.

Which categories of GO terms (BP, MF, CC) are shown in Ext Fig. 2b (especially the right one)? Overall, the presented data do not support the claims of the authors (and this is not a fate mapping approach). A comparison of EZ link pulldowns of sub-pleura vs interstitium would be more informative.

Why unchallenged lungs exhibit so much "de novo protein synthesis" (Fig. 2b)? Why, on the other hand, no new synthesis whatsoever of "pleural matrix" is observed? A zoom on pleura and mesothelial cells should be also provided.

In Fig. 2c the far right image is of higher magnification than the other two shown. Nevertheless, these images clearly show (within the methodological limitations) an overlap of de novo synthesized proteins and NHS-FITC labelled "premade" ones, in contrast with the authors claims.

ncAAS label only collagen? Why do the authors refer to NHS-ncAAS+ staining as newly synthesized collagen? Vice versa why do the authors refer to NHS+ncAAS- staining as "transferred" collagen, since all lysine containing proteins can be labelled? Btw, no claims on intracellular localization (e.g. cytoplasmic) can be made from these images; why does the anti-collagen antibody (specificity, negative controls?) stain everything? Please see comments on imaging above.

Imaging post AAV8 administration (Fig. 3b) is not sufficiently convincing, especially the lack of interstitial labeling. Please see comments on imaging above.

A consistent experimental setup should be employed across all methods and protocols, particularly in experiments involving mice. Deviations from the established setup should be explained, elucidating the rationale behind any modifications made to the experimental protocol. e.g. in Fig. 3 a/c AAV is administered either after (a) or before (c) NHS-FITC. If the results from the three reporters are to be shown in the same graph, t-test is not the statistical method of choice.

The choice of the selected publicly available scRNAseq datasets, among the plethora available (including integrated datasets), should be explained. Does the analysis of other datasets yield the same results? Basic information should also be provided: number of samples, number of cells analyzed. Mesothelial cells in the utilized dataset are very few for solid conclusions; which are the markers that discriminate/mark mesothelial cells in this dataset? Statistics of differential expression (Fig. 3e) should be provided; it's impossible to identify the expression pattern of the selected genes, as there are many more lines than the indicated genes.

The number, origin and preparation (fixed? OCT?) of human lung tissue samples should be provided. A negative control, i.e. healthy tissue, should also be used. The conclusions on matrix transfer from these experiments (Fig. 4) are not sufficiently convincing; "alveolar labeling" (Fig. 3b) is at background levels; passive diffusion of NHS should be somehow ruled out.

The "separation of pleura from deep interstitial tissues" (Fig. 4d) should be better explained. The experimental design should be better clarified. The process by which "Principal component analysis indicated that this NHS233 EZ-link+protein matrix was similar in composition to atrophic scars and to abnormal stiffened vascular connective tissue matrix" should be explained. PCA for GO enrichment? Please see also the relative points on GO analysis above. Moreover, statistics should always accompany differential expression results (Fig. 4g). "Fluidity factor" in extended Fig. 4 is not sufficiently substantiated; the introduction of novel terms should be very cautious.

The cell-cell and ligand-receptor analysis should be better introduced; statistics?

IL-18R imaging should include all the imaging controls, as prompted above, especially proof of colocalization. Moreover, most shown macrophages are interstitial and not alveolar! Cell counting was performed manually (and from how many reviewers) or with a software?

The experiment in Figure 3g should be better defined - by a graph maybe? What macrophages were they used for the transwell experiments? Were they activated and how? A short paragraph on IL-18 high AMs there belongs to the discussion.

Bleomycin-induced pulmonary fibrosis was only examined with histology and body weight (and the SMAD2/3 ratio), while the typical readout assays (collagen quantification, pulmonary edema, BALF inflammation, respiratory functions) were not used. Moreover, the histology images do not show much fibrosis; some histology images in Figure 7 are reminiscent of emphysema (or bad section preparation or inappropriate microscopic slides).

Reviewer #3

(Remarks to the Author)

In the publication entitled "Targeting pleuro-alveolar junctions resolve and reverse lung fibrosis" by Fischer and colleagues, the authors reveal new aspects of fibrosis pathomechanism. Activated alveolar macrophages (AMs) release IL-18, which stimulates mesothelial cells to produce Cathepsin B and enables the pleural matrix pool to be transferred deep into the lung interstitium, thereby contributing to fibrosis development. Blocking either IL-18 or Cathepsin proteolysis prevents the development of fibrosis and represents a possible therapeutic intervention. The data is interesting and of good quality. However, some issues require further attention and clarification.

- The NHS-EZ-LINK pull-down experiment described in Extended Figure 2 and later in Fig. 4 is interesting, but the experimental layout is unclear. As these results are central to the idea that a large fraction of ECM is premade and transferred during fibrosis, more details are necessary. What controls were used in the experiments? Did the authors include (bleomycin-treated mice that received) NHS without a biotin linker or (bleomycin-treated mice receiving) no labelling with NHS as controls? These controls are important to verify the specificity of the NHS-EZ-LINK approach.

-The histological images shown in Extended Fig. 1b (upper right) and 1e (PBS animal on the left) are the same, rotated 180°. It might be acceptable to show the controls twice when the experiments were performed together, but it raises suspicion, especially when the image is rotated and the experimental scheme shows different treatment protocols (1b: injection at day 0, analysis at day 14; 1e: injection at day -1, analysis at day 14).

-In respect to Fig. 3, can the authors show how strong the AAV8-dependent Col1-FLAG and Col2-FLAG overexpression was, e.g. real-time PCR? It would also be interesting to verify the results by using the nAAS system. Since Col1-FLAG and Col2-FLAG are ectopically expressed, their proteins should be newly synthesised (as mentioned by the authors) and accordingly marked in the presence of nAAS. This is different from the normal situation, in which a large part of the ECM is

premade.

-How were the human data represented in Fig. 4 controlled? The authors mentioned that patients with progressive interstitial lung disease were also included in the study. Can the authors include healthy or nonfibrotic controls? If not, then a control experiment with mouse explants is necessary to verify their approach (similar to what is shown in Fig. 5g): healthy as well as bleomycin-treated mouse explants should be isolated, surface NHS labeled, and the transfer of ECM into interior regions within 24h analysed. If this is bleomycin-dependent and not cell culture-dependent, then their human data can be correlated with mouse data. However, in the current form, the data may be an artifact of the in vitro culture system.

-In Fig. 5 and Fig. 6, the authors describe the expression of Il18 in published scRNA-Seq (Sikkema, L. et al. 2023). In this respect, they write in Figure legend 6 "a) UMAPs of cell populations in human IPF samples". However, upon closer inspection of the published data (<https://cellxgene.cziscience.com/e/9f222629-9e39-47d0-b83f-e08d610c7479.cxg/>), it appears that the UMAP depicted in Fig. 6A is derived from all patient samples, including healthy controls, and not IPF, as stated in the text. The data should be reanalyzed and adjusted for patients with fibrosis only. Furthermore, the quality of the histology shown in Fig. 5e and f is not completely sufficient. The staining pattern in Fig. 5e does not support the exclusive expression of IL-18 in hematopoietic cells. In addition, the strong CD45 and IL-18 signals in mouse tissue, as shown in Fig. 5f (upper left panel), suggest an unspecific staining pattern in the lung rather than a staining of hematopoietic cells. Can the authors provide a better analysis for this experiment (plus healthy control samples) and may even show an IL-18 flow cytometry, Il18 real-time PCR or ELISA for FACS-purified BAL AMs isolated from healthy as well as bleomycin-treated animals at different time points?

-The authors write on P. 11, l. 292 that they used IL-18^{high} AMs for coculture experiments. In the method, however, they describe the generation of GM-CSF-derived macrophages from monocytes. These cells are not AMs. Therefore the authors need to show that GM-CSF-derived cells express high levels of IL-18 in vitro.

Minor things

-It is easier for the reader if the authors could provide more methodological information in the Results section. For instance: P. 5, l.93: It would be helpful to mention in the text that NHS-FITC was administered intra-pleurally; P. 7, l. 149: Shortly mention here how mice were treated with nAAS; P. 7, l. 168ff: Please mention here that IFN γ -/- mice were used.

-P. 5, l.101: The authors wrote "We also confirmed that interstitial fibroblasts, immune cells and mesothelial cells were unlabelled with NHS-FITC in our system (Extended Fig. 1c)." Extended Fig. 1c does not show any evidence that immune cells are not labelled. PDGFR and M6A stainings are shown, which both do not label immune cells. And there seems to be some mislabeling in the figure legends for Extended Fig. 1c ("(d) e) Representative histology images of mouse livers.")

-P. 8, l. 198: It is unclear, why the authors analysed scRNA-Seq data (Fig. 3e) to examine collagen-related mesothelial gene expression kinetics during bleomycin-induced pulmonary fibrosis when they previously showed that the majority of ECM was not de novo synthesised but premade. If mesothelial cells express new collagen-related genes and secrete them to be incorporated into the ECM, as stated by the authors (P.9, l. 208), would you not be able to mark them with nAAS? Can the authors comment on this and add a respective sentence to the Results section (P. 8, l.198)?

-P.11, l.267: Please check the reference 40 mentioned here: "These findings are consistent with studies showing knockout of AMs impede bleomycin induced pulmonary fibrosis (40)." Ref 40 should be Zhang et al., 2018 (PMID: 30527805).

-Please check comma usage and writing throughout the text (comma usage for instance: abstract l. 43 ("into deep lung tissue occurs"), l. 47 ("Cystatin A shuts down"), l. 48 ("the pleura provides"); writing: p.5, l.107: "because it is in used"; Figure legend 2: "and three consecutive days nAAS were injected"). Check typos ("bleoymcin") and figure legends. Some mislabeling is evident (for instance, Extended Fig. 1d, as mentioned earlier, Fig. 4: b-c). Representative multiphoton images of NHS-FITC-labeled human lung tissues. Scale bars: a: 200 μ m and 100 μ m b: 100 μ m and 20 μ m). Also terms like "dramatic" are subjective and should be omitted in the results section.

Version 1:

Reviewer comments:

Reviewer #1

(Remarks to the Author)

The authors have appropriately responded to all of my concerns.

Reviewer #2

(Remarks to the Author)

Although the authors have tried to answer all comments from the reviewers, in some responses they did not address the essence of the question, and many suggested experiments, including animals or not, were not performed. Overall,

I remain unconvinced from imaging, where no metadata was added as requested.

I am still not convinced that the observed phenotype is not due to FHS leakage. I appreciate the adenoviral delivery of TGF (which puzzled me a lot). Still, the suggested Evans blue experiment, as well as the IV administration of BLM, would make a stronger case. Of course, "labeled matrix transfer only occurs when tissue damage occurs" as the authors state in their response, but this is when epi/endo/meso-thelial leakage also occurs.

I still think that the effect of Z-FA-FMK cannot be attributed to cathepsin B inhibition only.

I still don't grasp the concept of transferred ECM, and my relative question still stands: How is it possible for such a highly complex and linked macromolecular structure to move around? By which mechanisms?

Reviewer #3

(Remarks to the Author)

The authors answered all my questions sufficiently. I would like to congratulate the authors on this work. I would just recommend the deletion of the following words: p.4, l.82: "as well as M1 and M2 macrophages". The M1/M2 concept is outdated and should not be propagated.

Version 2:

Reviewer comments:

Reviewer #2

(Remarks to the Author)

I am afraid that I remain unconvinced.

PAJs remain poorly defined. No specific molecular markers or relevant publications have been included although requested.

Non-specific FHS leakage upon inflammation and fibrosis was not sufficiently ruled out. Controls from previous studies in healthy tissues are not and cannot be adequately convincing.

Imaging, that the paper relies upon, remains unconvincing. No further imaging-associated (meta)data have been included (e.g. orthogonal and k-curve analyses) as requested.

Therefore, the concept of mobile ECM remains not sufficiently convincing to me. How is it possible for a huge macromolecular structure to break all integrin-mediated cell attachments and migrate across barriers? Yes, proteolysis (and phagocytosis for that matter) could help but still ... How does mobile ECM integrate with the existing interstitial ECM?

RESPONSE TO REVIEWERS' COMMENTS

Reviewer #1 (Remarks to the Author):

This research is quite interesting and the findings are novel. These investigators explored the role of macrophages and found that these cells transfer connective tissue from pleuro-alveolar (extracellular matrix underneath the mesothelial lining) junctions to lung parenchyma. They found that pleuro-alveolar junction macromolecular transfer of connective tissue was present in human lung biopsies and chronic models of lung fibrosis, and was reduced in acute fibrosis models. Moreover, this process could be inhibited by Cystatin (cysteine protease inhibitor). However, several concerns need to be addressed by the investigators to improve the rigor and causality of the observations.

1. Imaging is not yet convincing. the investigators should provide more specifics (imaging parameters including exposures, antibodies used), the FITC channel has a lot of autofluorescence - how is distinguished from collagen/elastin, metadata should be included, and the human lung findings need to be better defined histologically.

Response: Thank you very much for your comments. Imaging exposure times ranged from 100-200 ms, depending on the slides and antibodies used. We added "Reagent or Resource" section in the revised manuscript. The antibodies are listed in the "**Reagent or Resource**" section, including the company, product ID, and working concentration.

Regarding the autofluorescence in immunofluorescence sections (IF), we have employed several methods to remove background and optimize the entire protocol both before and during staining:

1. Perfusing PBS before organ extraction to remove red blood cells.
2. Using Sudan Black B to reduce autofluorescence.
3. Performing endogenous tissue controls (without primary or secondary antibodies) and primary antibody controls (with only secondary antibody) to assess the levels of autofluorescence and non-specific binding in our IF experiments. Additionally, these control parameters were used to remove background intensity during statistical analysis.

To better define human lung samples histologically, we kindly provide two images of Masson's trichrome staining for IPF and peritumor human lung, 10x magnification, with a scale bar of 100 μm .

2. Specificity: Is this inward movement specific to ECM or would any abnormal protein or particle be similarly phagocytosed by macrophages and moved centrally?

Response: Thank you for your insightful comments. In our 2023 Nature Protocols paper, we described the use of NHS-FITC labeling in vivo. In brief, NHS esters form covalent amide bonds by reacting with primary amines at the N terminus and lysine residues of a polypeptide chain under slightly alkaline conditions (pH ~9). The chemical bond on proteins is irreversible, where any lysine residue is covalently tagged and therefore movement is specific to ECM proteins on lung serosal surfaces. In this study, we outline a protocol utilizing fluorophore-NHS ester-based ECM fate mapping to effectively tag nearly all ECM proteins on serosal surfaces, irreversibly. The small size of the moieties attached to the ECM proteins minimizes the risk of significant steric hindrance that could disrupt normal protein function. The covalently bound tags generated through our protocol remain stable for several weeks post-labeling and do not lead to increased recruitment of CD45+ immune cells or elevated pCaspase-dependent cell death. These findings strongly support the specificity of ECM protein movement.

3. Causality: The key observation is that macrophages are moving ECM from the pleura to the more central portions of the lung. While the investigators present convincing findings to support this finding, it's unclear whether the movement of ECM from the pleura to the central portions of the lung are responsible for the development of lung fibrosis. Additional experiments are needed to prove causality.

Response: Thank you very much for the comments. Based on our observations and as shown in Figure 6, blocking ECM movement, specifically, with intrapleural injection of inhibitors or viral delivery to lining mesothelium, both prevented the development of lung fibrosis. This clearly indicates a role for serosal ECM movement in the progression of fibrosis.

Due to current constraints/time need to obtain new animal experimental approval (estimated time of 6-8 months for any new approvals), we cannot perform additional animal experiments. We hope this explanation provides clarity on our findings, and we appreciate your understanding of the limitations we face.

Reviewer #2 (Remarks to the Author):

In this manuscript, Fischer et al. follow their own similar observations in skin and liver fibrosis on “mobile ECM” and suggest that mesothelial made ECM translocate to the interstitium (with unknown mechanisms) promoting pulmonary fibrosis. The authors suggest that the process is facilitated by macrophage derived IL-18 promoting cathepsin B expression from mesothelial cells. However, there are several conceptual as well as methodological issues that need to be addressed.

Major

1. Pleuro-alveolar junctions should be better defined (including some molecule markers, as well as hypothesized functions), as I could not find any relevant publications in the literature; the cited references are not relevant.

Response: Thank you very much for your comments. We have updated the related references. Additionally, we've also added a more detailed description to clarify the terminology of pleuro-alveolar junctions (PAJs). Specifically, we stated: “Pulmonary fibrotic diseases almost invariably lead to interstitial scarring and thickening of the connective tissue beneath the pleural membranes that encase the lung. These regions, termed as pleuro-alveolar junctions (PAJs), are where the pleural membranes interface with the alveolar interstitium. They are hypothesized to serve as sites of fibrotic remodelling that contribute to disease progression.” (Introduction part)

2. The intriguing concept of “ECM movement” should be better introduced and discussed. The previous publications from the same group, which decrease novelty, should be further discussed considering the findings here. Some questions should be entertained: how is it possible for such a highly complex and linked macromolecular structure to move around? By which mechanisms? Passive or energetic? In their previous publications suggested that fibroblasts can drug ECM cargo around. How?

Response: Thank you for your insightful comments. We agree that this mechanism needs further introduction and discussion. ECM movement is an energetic process rather than a passive one. In pulmonary fibrosis, AMs and the mesothelium at PAJs are key players. Activated by AMs, Cathepsin B breaks down connective tissue at PAJs, releasing ECM macromolecules that migrate into the lung interior to sustain fibrosis. Once Cathepsin B signaling is activated in the mesothelium, macrophages are no longer required for this process. Blocking ECM transfer at PAJs (with mesothelium-specific Cystatin A overexpression) prevents downstream fibroblast activation and matrix deposition, highlighting that targeting PAJ disassembly via the pleura provides a unique therapeutic avenue to resolve and reverse lung fibrotic diseases.

Our previous studies on skin and abdominal/peritoneal fibrosis first reported ECM macromolecule import in fibrosis. These findings indicate that ECM transfer is a general feature of tissue and organ healing, where ECM from the serosal lining contributes to fibrotic healing. In this study, we demonstrate that this mechanism is also active in the lungs, with ECM transfer and fibroblast activation together producing most of the fibrotic material. Although our earlier publications suggested that fibroblasts can move ECM cargo in the skin fascia, the mechanism in the lung involves different cellular and molecular players, reinforcing the novelty and significance of our current findings.

We have added in the discussion part to make it clearer: "In contrast to our previous findings in the fascia of the skin, where fibroblasts moved ECM cargo, the mechanism in the lung involves different cellular and molecular dynamics. Here, we reveal a novel mechanism in lung fibrosis where AMs disassemble and transfer ECM components from PAJs into deeper lung tissues through cysteine-type proteolysis, particularly involving Cathepsin B. This mesothelial activation is facilitated by communication with the AMs via the IL18 receptor. This energetic process indicates that signals at PAJs are essential for myofibroblast activation and maintains in deep lung tissue."

3. Although mesothelial cells and their ECM production has been reported to play a role in pulmonary fibrosis (mostly in humans), that should be cited btw, do mesothelial cells secrete ECM in the pleural cavity (besides the lung underneath) where NHS is administered? What for?

Response: Thank you for your comments. We have cited the relevant publications to support the function of mesothelial cells in pulmonary fibrosis, as shown below. Mesothelial cells produce and secrete various ECM components such as collagen, fibronectin, and proteoglycans including proteins with lubricant functions into the pleural space to circumvent pleural adhesions. Within the serosal surfaces, these components form a matrix that provides mechanical support and maintains the integrity of the pleural lining. They are involved in tissue repair and regeneration, as well as in inflammation and immune responses.

- 1 Sakai, T. et al. Myocardin regulates fibronectin expression and secretion from human pleural mesothelial cells. *Am J Physiol Lung Cell Mol Physiol* 326, L419-L430 (2024). <https://doi.org:10.1152/ajplung.00271.2023>
- 2 Yang, J. et al. Activation of calpain by renin-angiotensin system in pleural mesothelial cells mediates tuberculous pleural fibrosis. *Am J Physiol Lung Cell Mol Physiol* 311, L145-153 (2016). <https://doi.org:10.1152/ajplung.00348.2015>

4. NHS-FITC leakage upon the well-known bleomycin-induced increased endothelial/epithelial/mesothelial permeability is not shown or examined and could account for additional labelling post BLM. Intrapleural administration of evans blue followed by bleomycin should be performed to examine leakage; Ext Fig 1b does not

prove much.

Response: We thank the reviewer for the comment. We have repeatedly confirmed absence of leakage using NHS-ester in vivo. Please see Correa-gallegos et al 2019, 2023 and Fischer et al., 2022, 2023. In the absence of injury, transfer of labelled matrix is not observed, even after 40 days. Please see images below. At any given timepoint, labelled matrix transfer only occurs when tissue damage occurs. Further, we show that labeled matrix does not transfer in bleomycin treated animals subjected to mesothelium-specific Cystatin A over expression. We believe these experiments are evidence that passive movement or diffusion is not a source for this transfer.

5. In all imaging figures/panels, that the paper relies on, proof of claimed colocalizations (orthogonal and k-curve analyses) should be provided. The specificity of all antibodies should be provided (isotype and dilution controls). Higher/lower magnifications should also be provided; conclusions should not be based on one shown high magnification image; more optical fields, intensity quantifications and statistics are needed.

Response: We appreciate your feedback and have addressed your concerns in the revised manuscript. We have included detailed specific information for all antibodies, including isotype and dilution controls, which can be found in the Methods section and "Regent and Resource" section, with corresponding images provided below. We have added both higher and lower magnification images to the revised figures, offering robustness of our findings. We have demonstrated the consistency and reproducibility of the observed effects, as shown in our new updated **Figure 5b**. Finally, we have incorporated quantitative analyses of fluorescence intensities and detailed the statistical methods used in the Methods section. These revisions collectively enhance the robustness of our findings.

6. The choice of the herpes virus model is not explained sufficiently. Adenoviral delivery of TGF would be a much better choice, eliminating inflammatory contributions (and endo/meso-thelial leakage) to “ECM transfer”. Moreover, bleomycin effects are not fully analyzed and thus the drawn conclusions are questionable. Although IT/OA administration of bleomycin is the most frequent route of administration, the authors should administer BLM (also) intravenously (IV), since with this method fibrosis develops subpleurally (as in humans, and as opposed to perivascular development upon IT administration), and therefore much better for the purposes of this paper.

Response: Thank you very much for the input. According to your comment, we have now performed new in vivo TGF β and DN-TGF β RII over expression specifically into the lining mesothelium, using viral-based approaches. These new experiments have now been added as **extended figure 4** to the manuscript (see new image below). As the reviewer can see, over expressing TGF β in lining mesothelium, leads to matrix movement even in the absence of bleomycin instilment. Conversely, overexpressing a dominant negative form of TGF β receptors in lining mesothelium, blocks matrix movement, even in the presence of bleomycin instilment. These findings further support our conclusions that matrix movements are required for fibrotic development. Accordingly, the description and figure legends are added in the revised manuscript.

Extended Figure 4

Extended figure 4. TGFβ Induced Lung Fibrosis Model and Effects

(a) Workflow of TGFβ induced lung fibrosis model. Mice were intrapleurally injected with NHS-FITC labelling mix. The next day 100ng of recombinant TGFβ was injected, leading to increased mesothelial TGFβ signaling. Independent t-test was used for the comparison of two groups (n.s. = not significant, *** P<0.001). (b) Active mesothelial TGFβ signaling leads to matrix invasion 14 days post recombinant TGFβ injection (n = 5). Scale bars: 1000μM. (c) A single injection of recombinant TGFβ induced persistent active mesothelial TGFβ signaling and increased the number of M6A+ cells in lung interstitium. Scale bars: 50μM. Independent t-test was used for the comparison of two groups (*** P<0.001). (d) AAV based particles encoding for active TGFβ increase mesothelial TGFβ levels (n=3). Scale bars: 50μM. (e) Mesothelial TGFβ activation drives pulmonary fibrosis. AAV particles encoding for active TGF-β were applied intrapleurally (n=5). Scale bars: 1000μM. Independent t-test was used for the comparison of two groups (*** P<0.001). (f) Mesothelial TGFβ activation leads to increased interstitial pSMAD levels and M6A+ cells (n=5). Scale bars: 50μM. Independent t-test was used for the comparison of two groups (*** P<0.001). (g) AAV based particles encoding for dominant negative TGFβ receptors(AAV-DN-TGFβRII) express robustly in mesothelial cells (n=3). Scale bars: 50μM. (h) Targeted inhibition of TGFβ in mesothelial cells blocks bleomycin-induced invasion of fluid matrix. AAV particles encoding for dominant negative TGFβ receptors were applied intrapleurally; five days later NHS-FITC was installed intrapleurally; then one day later bleomycin was installed; 14 days after bleomycin installation organs were harvested (n=5). Scale bars: 1000μM. (i) Inhibition of mesothelial TGFβ blocked bleomycin-induced TGFβ signaling and increased the number of M6A+ cells in lung interstitium (n=5). Scale bars: 50μM. Independent t-test was used for the comparison of two groups (*** P<0.001). (j) Inhibition of mesothelial TGFβ prevents bleomycin-induced mortality (n=5). Log-rank test was used for statistical comparison. Data represented are mean ± SD.

We acknowledge the insightful comment regarding the administration of bleomycin in our study. We regret that we did not include the IV administration of BLM due to the lack of governmental approval for this method. However, we appreciate the suggestion and recognize its importance for studying fibrosis development subpleurally, mirroring the conditions in humans more accurately than perivascular development seen with IT administration. If given the opportunity in the future, we

are committed to exploring the IV administration of BLM. Thank you once again for your valuable feedback.

7. Results from the re-analysis of publicly available scRNAseq datasets should be mostly supplementary and not main figures (that should contain novel data). Moreover, the results of the re-analyses can only be used for hypothesis generation and further validation experiments are necessary to substantiate any claims.

Response: Thank you for your feedback. We have moved the re-analysis of public databases to the **extended figure 5** and **extended figure 7**. All main figures and conclusions are now based on novel data, and we have conducted additional experiments to validate our claims.

8. Does exposure of mesothelial cells (some established line or primary cells) to IL-18 promote Cathepsin B expression?

Response: Thank you for pointing this out. We cultured the Met5A cell line (human mesothelial cell line) with IL-18 pre-treatment at 10 ng/ml for 12 hours. We performed Western Blot using a Cathepsin B antibody. Our analysis showed that IL-18 enhances the expression of Cathepsin B on cultured mesothelial cells.

9. Z-FA-FMK is an inhibitor of not only Cathepsin B, but also cathepsins L, and S,

cruzain, and papain. Most importantly, Z-FA-FMK also selectively inhibits effector caspases 2, 3, 6, and 7.

Response: Thank you very much for your comment. We apologize for any confusion caused by the use of Z-FA-FMK. To address this issue and ensure accurate results, we performed a new ex vivo experiment to eliminate the potential impact of Z-FA-FMK's broad inhibitory effects. In this experiment, we co-cultured the lung punches with both Z-FA-FMK and a selective cathepsin B inhibitor (CA-074). The results remained consistent with our previous findings, as shown below.

10. Terminology (Pleuro-alveolar junctions, mobile ECM, fate mapping, fibroblast licensing, fluidity factor) should be used with caution.

Response: According to the reviewer's suggestions, we have explained or changed some of the terminology. As for the term "Pleuro-alveolar junctions", we added the explanation in the introduction part as "Pulmonary fibrotic diseases almost invariably lead to interstitial scarring and thickening of the connective tissue beneath the pleural membranes that encase the lung. These regions, termed as pleuro-alveolar junctions (PAJs), are where the pleural membranes interface with the alveolar interstitium. They are hypothesized to serve as sites of fibrotic remodelling that contribute to disease progression." In addition, we changed the term "mobile ECM" to "**transferred ECM**", and "fibroblast licensing" to "**fibroblast activation**", and "fluidity factor" to "**dynamic transfer**" in the new **extended Figure 6**. However, we believe that our terminology of fate mapping to the ECM, is appropriate in this context. As defined, fate mapping depicts the tracing of biological material (either cells, or proteins) in both spatial and temporal dimensions, showing how they move from their origins to their final locations and differentiate into specific tissues and structures over time. In our study, we used NHS-ester to trace the movement of serosal ECM across these dimensions, which aligns with the conventional definition of fate mapping. This approach allowed us to effectively capture the dynamic processes of ECM transfer across time and location.

Minor

11. The definition of macrophage subpopulations is too general. Alveolar vs interstitial, and resident vs inflammatory are very important distinctions, especially in the context of the paper. The statement “IL-18 expressing macrophages are the major immune cell population” is not sufficiently supported by the literature.

Response: Thank you for the feedback. As suggested by the reviewer, we have added a paragraph in the introduction part to better describe the macrophage subpopulations in lungs :” In idiopathic pulmonary fibrosis (IPF), macrophages exhibit diverse roles and distinct phenotypes depending on their location and activation state within the lung microenvironment. Macrophages can be classified based on location, origin, and immune phenotypes, including alveolar macrophages (AMs) and interstitial macrophages (IMs), monocyte-derived inflammatory macrophages (MDMs) and tissue-resident macrophages (TRMs), as well as M1 and M2 macrophages. Recent scRNAseq studies of human lung fibrosis have identified two categories of AMs: pro inflammatory and pro fibrotic. These macrophage subsets contribute to fibrosis by secreting cytokines, chemokines, and reactive oxygen species, which in turn promote fibroblast activation, collagen deposition, and tissue remodeling within the alveolar interstitium. Notably, AMs have been identified as significant contributors to the increased levels of Interleukin 18 (IL-18) and Interleukin 18 receptor alpha chain (IL-18R α) in the lungs of individuals with IPF.”

12. All cited references should be reevaluated: some are irrelevant, some are outdated, while some cited reviews are low impact publications; much more authoritative reviews exist.

Response: Thank you very much for your comments. We have screened 8 literature references for improvement. The following are detailed references (it was also marked in red in the revised manuscript).

8—Martinez, F. J. *et al.* Idiopathic pulmonary fibrosis. *Nat Rev Dis Primers* **3**, 17074 (2017). <https://doi.org:10.1038/nrdp.2017.74>

9—Richeldi, L., Collard, H. R. & Jones, M. G. Idiopathic pulmonary fibrosis. *Lancet* **389**, 1941-1952 (2017). [https://doi.org:10.1016/S0140-6736\(17\)30866-8](https://doi.org:10.1016/S0140-6736(17)30866-8)

11—Stancil, I. T. *et al.* Pulmonary fibrosis distal airway epithelia are dynamically and structurally dysfunctional. *Nat Commun* **12**, 4566 (2021). <https://doi.org:10.1038/s41467-021-24853-8>

12—Mei, Q., Liu, Z., Zuo, H., Yang, Z. & Qu, J. Idiopathic Pulmonary Fibrosis: An Update on Pathogenesis. *Front Pharmacol* **12**, 797292 (2021). <https://doi.org:10.3389/fphar.2021.797292>

19—Qian, W. *et al.* Complex Involvement of the Extracellular Matrix, Immune Effect, and Lipid Metabolism in the Development of Idiopathic Pulmonary Fibrosis. *Front Mol Biosci* **8**, 800747 (2021). <https://doi.org:10.3389/fmolb.2021.800747>

31—Jenkins, R. G. *et al.* An Official American Thoracic Society Workshop Report: Use of Animal Models for the Preclinical Assessment of Potential Therapies for Pulmonary Fibrosis. *Am J Respir Cell Mol Biol* **56**, 667-679 (2017).

<https://doi.org:10.1165/rcmb.2017-0096ST>

33—Strunz, M. *et al.* Alveolar regeneration through a Krt8+ transitional stem cell state that persists in human lung fibrosis. *Nat Commun* **11**, 3559 (2020).

<https://doi.org:10.1038/s41467-020-17358-3>

34—Schiller, H. B. *et al.* Time- and compartment-resolved proteome profiling of the extracellular niche in lung injury and repair. *Mol Syst Biol* **11**, 819 (2015).

<https://doi.org:10.15252/msb.20156123>

13. Second harmonic signal microscopy should be introduced. The SHG co-localization with the FITC signal should be proved. What does the suggested co-localization prove and why?

Response: Thank you very much for the input. We have added a short introduction into SHG: “Second-harmonic generation (SHG) microscopy has become a valuable method for examining fibrillar collagen within diverse tissue contexts. Leveraging its physical principles, SHG microscopy offers high sensitivity to the structural nuances of collagen fibrils and fibers, making it particularly adept at detecting pathological changes characteristic of conditions such as cancer, fibrosis, and connective tissue disorders.” In addition, we have proved in **Extended Figure 1b** co-localization of SHG with NHS labeling. The co-localization suggested that the SHG signal coincided with mature NHS-FITC+ collagen fibers in the lung plural, but also revealed additional structurally immature matrix filling gaps between adjacent mature collagen fibers.

14. Fibroblast identification in immunocytochemistry is not sufficiently convincing. Which PDGFR was used? PRFGRA and PDGRB mark distinct fibroblast subpopulations with opposing functions. Positive/negative controls, lower magnifications, proof of colocalization, intensity quantification and statistics are needed. The same applies to all imaging figures in the manuscript.

Response: thank you very much for the comments. We have confirmed that the antibody we used in the manuscript is PDGFR α (R&D systems; AF1062; working concentration 1:100) and we have changed it in all figures. In addition, we conducted new CD45 and α SMA staining for PBS-treated mice, demonstrating that labeling with FITC did not mark immune cells or myofibroblasts. Additionally, we observed colocalization of the PDGFR α antibody with Ai14xPDGFR α mice, confirming the utility of PDGFR α .

α SMA staining for PBS mice

15. There is discrepancy regarding BLM administration: the text mentions intratracheal injection (IT) and the methods oropharyngeal (OA).

Response: thank you very much for pointing it out. All the BLM administration is oropharyngeal (OA). We have changed all the related text accordingly to “oropharyngeal bleomycin installation” in the revised manuscript.

16. If NHS-FITC tagging post BLM is “linked to myofibroblast activation” how do the authors explain the lighting up of trachea and bronchi? Larger area images are also needed.

Response: We thank the reviewer for the comment. The images illustrated in Fig. 1f are taken in a light sheet microscope. In this setting tissue cleared BLM lungs were imaged to track the NHS-FITC+ matrix accumulation overtime. The first entry of NHS-FITC+ matrix into the lower lobes occurred on day 10 and subsequently the labelled matrix also infiltrated and thickened around the bronchi and bronchioles. These labelling is then captured in the microscope, thus providing evidence that matrix transfer indeed happened from the surface into the interstitial space. Higher magnification images are now provided in the figure as a subscript for better understanding of the moved matrix deposit near the bronchial space.

17. There is a discrepancy concerning Extended figure 1d-e between results' text and figure legend. Liver or lungs? Ext Fig. 1f should be better explained.

Response: thank you for pointing it out. We apologize for the confusion. The reason why we put the liver figure in Extended Fig. 1d is to show no detectable transfer of NHS-FITC+ matrix in other organs from bleomycin-treated animals, nor in labeled lungs without bleomycin-treatment, excluding random diffusion of NHS-FITC+-matrix in the animals. We have corrected the description in the figure legend of Extended Fig.1. Regarding Extended Fig. 1f, multiphoton microscopy was used to examine the surface and analyze the texture of the ECM through lacunarity values. A high lacunarity indicates a greater reduction in the NHS-FITC+ matrix with larger gaps, consistent with ECM being liberated from serosal surfaces in mice treated intrapleurally with bleomycin.

18. Which categories of GO terms (BP, MF, CC) are shown in Ext Fig. 2b (especially the right one)? Overall, the presented data do not support the claims of the authors (and this is not a fate mapping approach). A comparison of EZ link pulldowns of sub-pleura vs interstitium would be more informative.

Response: thank you very much for pointing it out. We have changed the Extended Fig.2b and added comprehensive results of GO term analysis in the result section: “The proteomics analysis revealed that 52% of all transferred extracellular matrix

proteins belong to ECM glycoproteins, with collagens accounting for 36%, ECM-affiliated proteins for 8%, and proteoglycans for 4% (Extended Fig. 2b). Notably, the identified collagens include members from both the fibrillary (Col1a1, Col1a2, Col3a1) and basement membrane (Col4a1, Col4a2) collagen families (Extended Fig. 2c). GO biological process analysis indicated enrichment primarily in extracellular matrix organization, extracellular structure organization, and external encapsulating structure organization. In terms of cellular components, significant enrichment was observed in collagen-containing extracellular matrix, basement membrane, and endoplasmic reticulum lumen. GO molecular function analysis showed enrichment in platelet-derived growth factor binding, protease binding, and phospholipase inhibitor activity. Phenotype analysis suggested that the transferred matrix resembles scar tissue in humans (Extended Fig. 2d)."

In addition, we agree that such a comparison would provide more informative insights. Consequently, in Figure 4 of our study on human lung proteomics, we have included a detailed comparison between the pleura and interstitium, which enhances our understanding of the differential protein expression and ECM composition in these regions.

19. Why unchallenged lungs exhibit so much “de novo protein synthesis” (Fig. 2b)? Why, on the other hand, no new synthesis whatsoever of “pleural matrix” is observed? A zoom on pleura and mesothelial cells should be also provided.

Response: We thank the reviewer for the comment. ECM is a structural component needed to maintain any organ in its normal active form for proper functioning. Constant wear and tear happen in the organs daily to maintain proper function. ECM maintains the elasticity of an organ allowing cells to function and communicate among each other for sustenance. The images in fig.2b are an example of how this homeostasis is maintained overtime and what changes occur during an injury like the onset of fibrosis. The newly synthesized fibers are short and without a given orientation simply due to normal progression of organ functioning. A minimum synthesis of collagen is observed in healthy lungs largely in cell cytoplasm, but not found drastically in the extracellular space. As seen in Fig. 2c, the onset of fibrosis is marked by increased amount of newly synthesized proteins that are released extracellularly to contribute to thick ECM fibers in the alveolar regions.

There is also a similar base level production of newly synthesized protein in the pleural region. This can be clearly seen in the 20x high magnification images (shown below).

20. In Fig. 2c the far right image is of higher magnification than the other two shown. Nevertheless, these images clearly show (within the methodological limitations) an overlap of de novo synthesized proteins and NHS-FITC labelled “premade” ones, in contrast with the authors claims.

Response: Thank you for your comment. We apologize for any confusion caused by the images in the previous figure. In the initial submission, our intention was to show a higher magnification of the white square area. However, the presentation might have led to misunderstandings. To address this, we have corrected the images. In the revised figure, we have ensured that the merged image and the individual channel images are shown at the same magnification, providing a more accurate and consistent representation of the data, as shown below. By using image J to inform the percentage of co-localized signals, we found that only around 13% signals were overlapped between the individual channels out of the total signals coming from the channels individually.

21. ncAAS label only collagen? Why do the authors refer to NHS-ncAAS+ staining as newly synthesized collagen? Vice versa why do the authors refer to NHS+ncAAS-staining as “transferred” collagen, since all lysine containing proteins can be labelled? Btw, no claims on intracellular localization (e.g. cytoplasmic) can be made from these images; why does the anti-collagen antibody (specificity, negative

controls?) stain everything? Please see comments on imaging above.

Response: We thank the reviewer for the comment. ncAAS is a method to label newly synthesized proteins throughout bleomycin treatment, with a fluorescent tag. NHS-FITC pleural labelling was performed to label only ECM components on serosal surfaces alone with a lysine residue at a given timepoint (Fig.2a). This ensured that all the already produced ECM had a fluorescent FITC tag. The ncAAS injections were only given subsequently (Fig.2a) to label any ECM protein that was synthesized thereafter. This made sure that we could distinguish premade serosal ECM (which is NHS+and ncAAS-) vs newly synthesized ECM proteins (which is NHS- and ncAAS+). We also wanted to precisely trace which component of ECM are being transferred (i.e. NHS+ncAAS-) and this method allowed us to determine this spatially. We have provided high magnification images to clearly show cytoplasmic build-up of newly synthesized proteins that also do not contribute to the extracellular space.

The anti-collagen antibody is a Pan-collagen that labels Col I,II and VI, thus covering a majority of what ECM is made up of. Therefore, a staining with this antibody leads to a general overall distribution of fluorescence signal in and around the tissue. The detailed information of the antibody has been added in the “Reagent or Resource” section.

22. Imaging post AAV8 administration (Fig. 3b) is not sufficiently convincing, especially the lack of interstitial labeling. Please see comments on imaging above.

Response: thanks for the comments. In Figure 3, we employed an AAV8-based system to introduce collagen 1-FLAG (Col1-FLAG), collagen 2-FLAG (Col2-FLAG), and a collagen-helix binding CNA35-mCherry reporter tag specifically into mesothelial cells. The transduced cells that initiate the synthesis of collagen transcripts produce Col1-FLAG, Col2-FLAG, or CNA35-mCherry fusion proteins. These proteins, upon integration into fresh collagen helices, allow identification of the cell or tissue origins for newly deposited collagen. Therefore, as we addressed in the manuscript “Intraleural administration of AAV8-Col1/2-FLAG reporter resulted in a specific viral transduction of the mesothelial monolayer, without any interstitial labeling”.

23. A consistent experimental setup should be employed across all methods and protocols, particularly in experiments involving mice. Deviations from the established setup should be explained, elucidating the rationale behind any modifications made to the experimental protocol. e.g. in Fig. 3 a/c AAV is administered either after (a) or before (c) NHS-FITC. If the results from the three reporters are to be shown in the same graph, t-test is not the statistical method of choice.

Response: thanks for your feedback.

(1)The reason that label NHS-FITC before AAV administered is that the pre-existing ECM was initially labeled with NHS-FITC, followed by the administration of AAV to ascertain whether the mobility indeed originates from mesothelial cells.

(2)The rationale behind administering AAV before NHS-FITC is rooted in the timeframe for AAV expression, typically occurring within 2-7 days. Hence, we injected AAV 5 days prior to NHS-FITC injection to see if simultaneous transfer of both pre-existing ECM and newly synthesized ECM from the pleura surface into lung tissue.

For Fig3c, we used multiple unpaired t tests to compare the control group and the bleomycin group within each reporter category. This is appropriate as there are only two groups being compared for each reporter. The analysis did not involve comparing the three reporters against each other, which would have required ANOVA. In addition, we have checked the statistical methods for all the experiments in the manuscript and made corrections accordingly.

24. The choice of the selected publicly available scRNAseq datasets, among the plethora available (including integrated datasets), should be explained. Does the analysis of other datasets yield the same results? Basic information should also be provided: number of samples, number of cells analyzed. Mesothelial cells in the utilized dataset are very few for solid conclusions; which are the markers that discriminate/mark mesothelial cells in this dataset? Statistics of differential expression (Fig. 3e) should be provided; it's impossible to identify the expression pattern of the selected genes, as there are many more lines than the indicated genes.

Response: thank you very much for the comment. We selected this publicly available scRNA-seq dataset due to its high data quality, comprehensive coverage of cell types, and relevance to our research question. As described in Figure 3, we aimed to determine whether the transferred matrix indeed originates from mesothelial cells. To achieve this, we analyzed scRNA-Seq data to understand the dynamic regulation of collagen-related genes in mesothelial cells during fibrosis. This analysis could reveal the potential role of mesothelial cells in responding to fibrotic signals and their contribution to the remodeling process. The dataset includes 28 samples and 29,297 cells, with 652 mesothelial cells, which is within the normal range. We have added this information in the Methods part. MSLN (Mesothelin) and UPK3B (Uroplakin 3B) were chosen as markers for mesothelial cells, as shown below. Figure 3e(new extended figure 5a) shows the z-scores of gene expression, calculated by subtracting the mean from each individual data point and dividing by the standard deviation. We apologize for any confusion caused. In Figure 3e (new extended Figure 5a), we have selected representative genes based on their relevance and significance as discussed in our manuscript. Due to space constraints and the complexity of the data, it is not feasible to display all genes in the figure. So we chose a subset of genes that effectively illustrate the expression patterns and

trends observed in our study.

25. The number, origin and preparation (fixed? OCT?) of human lung tissue samples should be provided. A negative control, i.e. healthy tissue, should also be used. The conclusions on matrix transfer from these experiments (Fig. 4) are not sufficiently convincing; “alveolar labeling” (Fig. 3b) is at background levels; passive diffusion of NHS should be somehow ruled out.

Response: Information about human lung tissue samples was added in the method part as “Peritumor human lung tissues(n=5) and IPF human lung tissues(n=5) were obtained from hospital, and then labelled by mixing NHS-ester 1:1 with 100 mM pH

9.0 sodium bicarbonate buffer and cultivated for 24 hours as described above”. Additionally, peritumor human lung tissue is added in figure 4b as negative control.

To rule out passive diffusion, we performed additional experiments, as shown below. As shown in (a), in the first group (control), we labeled the tissue using our standard protocol. In the second group, we used Tris to deactivate NHS-FITC before labeling the tissue. In the third group, we fixed the tissue with PFA overnight and then labeled it subsequently. As shown in Figure (a), both the Tris-deactivated and PFA-fixed groups exhibited no significant movement compared to the control group. This indicates that passive diffusion is unlikely to be responsible for the observed labeling. We further validated these findings with in vivo experiments, as shown in Figure (b). We compared the results at Days 1, 14, and 45 in a healthy mouse model. There were no significant changes over time, supporting our conclusion that the process is not due to passive diffusion. In Figure (c), we cultured lung tissue with and without the protease inhibitor. The Western Blot results clearly show a difference between the two groups. This suggests that the process is active, rather than passive diffusion.

26. The “separation of pleura from deep interstitial tissues” (Fig. 4d) should be better explained. The experimental design should be better clarified. The process by which “Principal component analysis indicated that this NHS233 EZ-link+protein matrix was similar in composition to atrophic scars and to abnormal stiffened vascular connective tissue matrix” should be explained. PCA for GO enrichment? Please see also the relative points on GO analysis above. Moreover, statistics should always accompany differential expression results (Fig. 4g). “Fluidity factor” in extended Fig. 4 is not sufficiently substantiated; the introduction of novel terms should be very

cautious.

Response: Thank you very much for the comment. We have added more details on the “separation of pleura from deep interstitial tissues”. In the method section, we added the following text: “To separate the pleura from deep interstitial tissues, firstly, the tissue sample was carefully isolated to expose the pleural surface. Subsequently, gentle traction and blunt dissection was applied to gradually separate the pleura from the underlying interstitial tissues, taking care to avoid inadvertent damage to either structure.” which is consistent with the scheme.

In addition, GO analysis in Fig 4 has been updated: Among all transferred ECM proteins, approximately 30% are ECM glycoproteins, 22.28% are ECM regulators, and 15.84% are collagens (Fig. 4e). GO term analysis revealed significant enrichment in biological processes such as extracellular matrix organization, external encapsulating structure organization, and extracellular structure organization. In terms of cellular components, the most enriched categories were collagen-containing extracellular matrix, endoplasmic reticulum lumen, and intracellular organelle lumen. Molecular function analysis highlighted enrichment in endopeptidase inhibitor activity, protease binding, and serine-type endopeptidase inhibitor activity. Phenotype analysis indicated that the NHS-EZ-link+protein matrix resembled atrophic scars in humans (Fig. 4f).

Figure 4

a

c

d

e

f GO Biological Process

g

After careful consideration, “fluidity factor” was deleted and changed to “dynamic transfer” to better explain the dynamic movement.

Extended Figure 4

27. The cell-cell and ligand-receptor analysis should be better introduced; statistics?

Response: Thank you for pointing this out. According to the author's recommendation, Cell-cell interaction was performed using the CellChat package (Suoqin Jin et al., CellChat for systematic analysis of cell-cell communication from single-cell and spatially resolved transcriptomics, bioRxiv 2023), which addressed in the Method part. P value less the 0.05 was considered as significant and standard analysis procedure was performed.

28. IL-18R imaging should include all the imaging controls, as prompted above, especially proof of colocalization. Moreover, most shown macrophages are interstitial and not alveolar! Cell counting was performed manually (and from how many reviewers) or with a software?

Response: Thank you very much for your feedback. For IL-18R imaging controls and colocalization, we have included all relevant imaging controls as previously mentioned. These additional data are presented in the revised manuscript. In addition, we apologize for the confusion caused, and we acknowledge the distinction between interstitial and alveolar macrophages. To clarify, we have performed additional staining using CD11c, a specific marker for alveolar macrophages in both human and mouse tissues. This new data is included in the revised manuscript (Figure 5). Lastly, automated cell counting was performed by Fiji (version 2.14.0).

29. The experiment in Figure 3g should be better defined - by a graph maybe? What macrophages were they used for the transwell experiments? Were they activated

and how? A short paragraph on IL-18 high AMs there belongs to the discussion.

Response: Thank you for pointing this out. We agree that it would be clearer to describe this in vitro experiment through a flowchart graph. Therefore, we modified Figure 5 and added a new flow chart. In the transwell assay, the BMDMs were cultured with RPMI 1640 (GIBCO) supplemented with 10% (v/v) FBS (Sigma), 20ng/ml GM-CSF (Sigma), 1% (v/v) Pen-strep (Sigma), and 1% (v/v) sodium pyruvate (GIBCO) 7 days for differentiation. Then the macrophages were collected for purification of IL-18+ macrophages by using MACS sorting beads. The method “Ex vivo culture of lung biopsies” and “Monocyte isolation and differentiation” were modified accordingly.

Additionally, we added descriptions about IL-18+ macrophages in the results section: " Moreover, culturing lung biopsies with recombinant IL-18 induced matrix transfer from PAJs. We used GM-CSF treated bone marrow derived macrophages (BMDMs), which are known to act as a proinflammatory cytokine that enhances antigen presentation and drives macrophages into a proinflammatory phenotype that produces inflammatory cytokines. We found that transwell co-cultures of lung biopsies with IL-18+ macrophages also induced significant transfer of NHS+ matrix from PAJs. Consistent with the crucial role of IL-18, a blocking antibody against IL-18 prevented this matrix transfer (Fig. 5e and f). "

30. Bleomycin-induced pulmonary fibrosis was only examined with histology and body weight (and the SMAD2/3 ratio), while the typical readout assays (collagen quantification, pulmonary edema, BALF inflammation, respiratory functions) were

not used. Moreover, the histology images do not show much fibrosis; some histology images in Figure 7 are reminiscent of emphysema (or bad section preparation or inappropriate microscopic slides).

Response: Thanks for the valuable suggestion. As suggested by the reviewer, we have added α SMA and pan collagen staining for the different timepoint of bleomycin-induced pulmonary fibrosis, please find the results shown below. In the immunofluorescence staining of α SMA and pan collagen, we could find that from day 3 to day 45, every time point was statistically different from the control group. Especially on day 14, the P value of α SMA was 0.0004 and that of pan collagen was less than 0.0001.

Reviewer #3 (Remarks to the Author):

In the publication entitled “Targeting pleuro-alveolar junctions resolve and reverse lung fibrosis” by Fischer and colleagues, the authors reveal new aspects of fibrosis pathomechanism. Activated alveolar macrophages (AMs) release IL-18, which stimulates mesothelial cells to produce Cathepsin B and enables the pleural matrix pool to be transferred deep into the lung interstitium, thereby contributing to fibrosis development. Blocking either IL-18 or Cathepsin proteolysis prevents the development of fibrosis and represents a possible therapeutic intervention. The data is interesting and of good quality. However, some issues require further attention and clarification.

1. - The NHS-EZ-LINK pull-down experiment described in Extended Figure 2 and later in Fig. 4 is interesting, but the experimental layout is unclear. As these results are central to the idea that a large fraction of ECM is premade and transferred during fibrosis, more details are necessary. What controls were used in the experiments? Did the authors include (bleomycin-treated mice that received) NHS without a biotin linker or (bleomycin-treated mice receiving) no labelling with NHS as controls? These controls are important to verify the specificity of the NHS-EZ-LINK approach.

Response: thank you for your insightful review. We acknowledge the importance of controls in experiments. In our setup, we used Dynabeads M-270 Streptavidin to pull down only the proteins labeled with NHS-EZ-LINK, which were then detected in the pleura and interstitium using proteomics. We have revised the methods section to provide clearer details: “For further MS analysis, protein pulldown was performed as follows, which is only to extract proteins labeled with NHS-EZ-LINK from both pleura and interstitium.”

2. -The histological images shown in Extended Fig. 1b (upper right) and 1e (PBS animal on the left) are the same, rotated 180 °. It might be acceptable to show the controls twice when the experiments were performed together, but it raises suspicion, especially when the image is rotated and the experimental scheme shows different treatment protocols (1b: injection at day 0, analysis at day 14; 1e: injection at day -1, analysis at day 14).

Response: thank you very much for the suggestions. We apologize for the confusion, and we have changed the figure to avoid using repetitive controls in the revised Extended Fig. 1b and 1e.

3. -In respect to Fig. 3, can the authors show how strong the AAV8-dependent Col1-FLAG and Col2-FLAG overexpression was, e.g. real-time PCR? It would also be interesting to verify the results by using the nAAS system. Since Col1-FLAG and Col2-FLAG are ectopically expressed, their proteins should be newly synthesised (as mentioned by the authors) and accordingly marked in the presence of nAAS. This is different from the normal situation, in which a large part of the ECM is premade.

Response: We thank the reviewer for the comment. Generating new real-time PCR measurements of bleomycin instilled lungs, would require new animal experiments for which we currently do not have permissions (estimated time of 6-8 months for any new approvals) and is beyond a reasonable timeframe for a revision. In Figure 3a, we transduced mesothelial cells with AAVs to label newly synthesized Collagen I and Collagen II with a fluorescent reporter that was driven by a mesothelial promoter. The aim of the experiment was to study mesothelial cell driven collagen deposition under fibrotic stimulation and not to overexpress Col I & II. The ncAAS system on the other hand, introduces a fluorescent reporter into any new protein that has been synthesized upon injection to distinguish between premade and newly formed protein. This is a broad labelling system and does not answer our question regarding the cell type responsible for pleural matrix deposition that is transferred in the interstitial space upon injury. The Col1-FLAG and Col2-FLAG proteins are co-expressed with NHS-FITC and are transferred together upon injury. Both these proteins are premade on the day of injury, and they move towards the interstitial space together. In healthy lung, these Col1-FLAG and Col2-FLAG proteins would be homeostatically produced and expressed.

4. -How were the human data represented in Fig. 4 controlled? The authors mentioned that patients with progressive interstitial lung disease were also included in the study. Can the authors include healthy or nonfibrotic controls? If not, then a control experiment with mouse explants is necessary to verify their approach (similar to what is shown in Fig. 5g): healthy as well as bleomycin-treated mouse explants should be isolated, surface NHS labeled, and the transfer of ECM into interior regions within 24h analysed. If this is bleomycin-dependent and not cell culture-dependent, then their human data can be correlated with mouse data. However, in the current form, the data may be an artifact of the in vitro culture system.

Response: thank you very much for pointing it out. In the new Figure 4, we have added the peritumor human lung tissue as nonfibrotic negative control, as shown below.

Figure 4

5. -In Fig. 5 and Fig. 6, the authors describe the expression of IL18 in published scRNA-Seq (Sikkema, L. et al. 2023). In this respect, they write in Figure legend 6 “a) UMAPs of cell populations in human IPF samples”. However, upon closer inspection of the published data (<https://cellxgene.cziscience.com/e/9f222629-9e39-47d0-b83f-e08d610c7479.cxg/>), it appears that the UMAP depicted in Fig. 6A is derived from all patient samples, including healthy controls, and not IPF, as stated in the text. The data should be reanalyzed and adjusted for patients with fibrosis only.

Response: Thank you very much for your insightful feedback. We apologize for the discrepancy between the figure legend and the data presented. As per your suggestion, we have reanalyzed the data, focusing exclusively on patients with fibrosis. The UMAP depicted in new Extended figure 7A now reflects only the samples from fibrotic patients. Cell-cell interaction is only containing fibrosis data. Please find the updated figures below.

6. Furthermore, the quality of the histology shown in Fig. 5e and f is not completely

sufficient. The staining pattern in Fig. 5e does not support the exclusive expression of IL-18 in hematopoietic cells. In addition, the strong CD45 and IL-18 signals in mouse tissue, as shown in Fig. 5f (upper left panel), suggest an unspecific staining pattern in the lung rather than a staining of hematopoietic cells. Can the authors provide a better analysis for this experiment (plus healthy control samples) and may even show an IL-18 flow cytometry, IL18 real-time PCR or ELISA for FACS-purified BAL AMs isolated from healthy as well as bleomycin-treated animals at different time points?

Response: Thank you for your feedback. We have improved the quality of the images in the revised Figure 5. We have refined our staining protocols, particularly for the expression of IL-18R and IL-18 (see in the revised Figure 5a and c). We showed both 10x and 20x magnifications in the revised Figure 5b, and additionally we included healthy mouse tissue and human peritumor samples to validate our findings in the revised Figure 5d. We also repeated the CD45 staining to confirm its specific staining pattern. These changes are shown in the revised manuscript.

derived macrophages from monocytes. These cells are not AMs. Therefore the authors need to show that GM-CSF-derived cells express high levels of IL-18 in vitro.

Response: thank you very much for your feedback. We have addressed this concern by performing additional experiments. Specifically, we conducted qPCR analysis to confirm that the GM-CSF-derived macrophages used in our coculture experiments express high levels of IL-18. The results clearly show that these cells indeed exhibit elevated IL-18 expression, validating their use as IL-18 high AMs in our study.

Minor things

8. -It is easier for the reader if the authors could provide more methodological information in the Results section. For instance: P. 5, l.93: It would be helpful to mention in the text that NHS-FITC was administered intra-pleurally; P. 7, l. 149: Shortly mention here how mice were treated with ncAAS; P. 7, l. 168ff: Please mention here that IFN γ ^{-/-} mice were used.

Response: thank you for your valuable feedback. We have made the requested corrections in the revised manuscript to provide more information in the Results section. Specifically, we have added that “By using N-hydroxysuccinimide ester fluorescein isothiocyanate (NHS-FITC) intrapleurally to tag the matrix we could follow any matrix transfer in real time in live mice”, “Wild-type mice were daily given ncAAS intraperitoneally for three days”, and “we opted for a viral-based mouse model in which intra-nasal herpes virus is instilled into IFN- γ -R^{-/-} mice leading to pneumonia”.

9. -P. 5, l.101: The authors wrote “We also confirmed that interstitial fibroblasts,

immune cells and mesothelial cells were unlabelled with NHS-FITC in our system (Extended Fig. 1c).” Extended Fig. 1c does not show any evidence that immune cells are not labelled. PDGFR and M6A stainings are shown, which both do not label immune cells. And there seems to be some mislabeling in the figure legends for Extended Fig. 1c (“(d) e) Representative histology images of mouse livers.”)

Response: thank you very much for pointing it out. We apologize for the confusion and we have added CD45 staining image in the revised Extended fig 1c to show that immune cells are not labelled (as shown below). Also, we have changed the mislabeling in the figures: “d) Representative histology images of mouse lungs treated with PBS and livers treated with bleomycin. e) Representative histology images of mouse lungs”. In addition, we double checked all the figures and legends to avoid any mistakes.

10. -P. 8, l. 198: It is unclear, why the authors analysed scRNA-Seq data (Fig. 3e) to examine collagen-related mesothelial gene expression kinetics during bleomycin-induced pulmonary fibrosis when they previously showed that the majority of ECM was not de novo synthesised but pre-made. If mesothelial cells express new collagen-related genes and secrete them to be incorporated into the ECM, as stated by the authors (P.9, l. 208), would you not be able to mark them with nCAAS? Can the authors comment on this and add a respective sentence to the Results section (P. 8, l.198)?

Response: Thank you very much for the insightful comment. As described in Figure

3, we aimed to determine whether the transferred matrix indeed originates from mesothelial cells. To achieve this, we analyzed scRNA-Seq data to understand the dynamic regulation of collagen-related genes in mesothelial cells during fibrosis. This analysis could reveal the potential role of mesothelial cells in responding to fibrotic signals and their contribution to the remodeling process. In addition to transferred ECM, new ECM is constantly being produced by mesothelium during fibrosis, replacing transferred ECM with new ECM at serosal surfaces.

Although the bulk of ECM components are pre-made, mesothelial cells upregulate collagen-related genes to support ECM turnover or modification during fibrosis progression. Incorporating scRNA-Seq analysis allows us to capture the gene expression kinetics and the contribution of mesothelial cells.

As mesothelial cells express new collagen-related genes and secrete them to be incorporated into the ECM, this process can be tracked using ncAAS. However, scRNA-Seq offers a broader and more comprehensive view of cellular responses and gene expression changes that ncAAS alone might not fully capture, particularly in terms of cell-type-specific expression. We have added a sentence to the Results section to clarify this rationale: "The scRNA-Seq analysis was performed to capture the gene expression dynamics in mesothelial cells, providing insights into their potential role in ECM remodeling during fibrosis, complementing the observations from ncAAS tagging."

11. -P.11, l.267: Please check the reference 40 mentioned here: "These findings are consistent with studies showing knockout of AMs impede bleomycin induced pulmonary fibrosis (40)." Ref 40 should be Zhang et al., 2018 (PMID: 30527805).

Response: thank you for recommending this article (Zhang et al.,2018), which proves the important role of IL18 in pulmonary fibrosis. We have cited this, and we have also cited a reference supporting the role of alveolar macrophages in pulmonary fibrosis(Byrne, A. J., Maher, T. M. & Lloyd, C. M. Pulmonary Macrophages: A New Therapeutic Pathway in Fibrosing Lung Disease? Trends Mol Med 22, 303-316 (2016). <https://doi.org/10.1016/j.molmed.2016.02.004>)

12. -Please check comma usage and writing throughout the text (comma usage for instance: abstract l. 43 ("into deep lung tissue occurs"), l. 47 ("Cystatin A shuts down"), l. 48 ("the pleura provides"); writing: p.5, l.107: "because it is in used"; Figure legend 2: "and three consecutive days ncAAS were injected"). Check typos ("bleoymcin") and figure legends. Some mislabeling is evident (for instance, Extended Fig. 1d, as mentioned earlier, Fig. 4: b-c). Representative multiphoton images of NHS-FITC-labeled human lung tissues. Scale bars: a: 200 μm and 100 μm b: 100μm and 20 μm). Also terms like "dramatic" are subjective and should be omitted in the results section.

Response: Thank you for your detailed review and valuable feedback. We have

carefully revised the entire manuscript, addressing all concerns raised including comma usage, writing clarity, typos, figure legends, and mislabeling issues. These revisions have been made to enhance the overall quality, accuracy, and clarity of our work. We appreciate your suggestions and believe these changes significantly improve the manuscript.

RESPONSE TO REVIEWERS' COMMENTS

Reviewer #2 (Remarks to the Author):

Although the authors have tried to answer all comments from the reviewers, in some responses they did not address the essence of the question, and many suggested experiments, including animals or not, were not performed. Overall, I remain unconvinced from imaging, where no metadata was added as requested.

Response: New animal experiments are included in the revised manuscript using mesothelium-specific viral over expression/down-regulation of TGFbeta signaling and its effects on lung fibrosis (see extended fig 4). We have also uploaded the metadata for the revised figures as supplementary material in the google drive:

(https://drive.google.com/file/d/1KU9CN2ERSJsRVz54prSXJSp8yiU5-dl5/view?usp=drive_link.)

I am still not convinced that the observed phenotype is not due to FHS leakage. I appreciate the adenoviral delivery of TGF (which puzzled me a lot). Still, the suggested evans blue experiment, as well as the IV administration of BLM, would make a stronger case. Of course, "labeled matrix transfer only occurs when tissue damage occurs" as the authors state in their response, but this is when epi/endo/meso-thelial leakage also occurs.

Response: We appreciate the reviewer's thoughtful comments and suggestions regarding potential NHS leakage. We have repeatedly confirmed the absence of leakage using NHS-ester in vivo across several studies (Correa-gallegos et al Nature 2019, Nature 2023; Fischer et al., Nature Communications 2022). As detailed in our published protocol (Fischer et al., Nature Protocols 2023), we have extensively validated the use of NHS-ester labeling to study ECM dynamics in internal organs and wound healing. This method allows specific and stable labeling of ECM proteins, with signal retained on fibrous ECM structures but does not trigger increased recruitment of CD45+ immune cells or increased pCaspase-dependent cell death.

That said, we acknowledge the value of the suggested Evans blue experiment and IV administration of bleomycin to further verify our results. These are excellent suggestions that could provide additional evidence to support our conclusions. However, due to current constraints and the time required to obtain new animal experimental approvals (estimated 6-8 months for any new protocols), we are unable to perform additional animal experiments at this time. Given these constraints, we have added a discussion of these potential follow-up experiments as a limitation in our paper. We will explicitly note that future

studies using Evans blue or IV bleomycin administration could further validate the specificity of the matrix transfer we observed. This will acknowledge the reviewer's insightful suggestions while being transparent about the current limitations. We hope this approach addresses the reviewer's concerns to the extent possible given our current constraints.

I still think that the effect of Z-FA-FMK cannot be attributed to cathepsin B inhibition only.

Response: We appreciate the reviewer's comment. We apologize again for any confusion caused by the use of Z-FA-FMK. Z-FA-FMK is a broad-spectrum cysteine protease inhibitor, not exclusive to cathepsin B. We agree this is a limitation of our study. We will address this in our discussion section, acknowledging that the effects of Z-FA-FMK may not be solely due to cathepsin B inhibition but could involve other cysteine proteases. However, we would like to emphasize that we have taken additional steps to verify the specific role of cathepsin B. As shown in Figure 6h-l (see below), we used AAV-mediated overexpression of cathepsin B, specifically, and its inhibitor cystatin A to further validate cathepsin B's role in vivo.

Based on the two points mentioned above, we added the limitation in the discussion part as below, highlighting in red in the revised manuscript:

“While our study provides valuable insights into ECM dynamics and protease involvement, we acknowledge certain limitations. Firstly, although we have extensively validated the NHS-ester labeling method for studying ECM in vivo, additional experiments such as Evans blue assays or bleomycin IV

administration could further confirm the specificity of the observed matrix transfer. Secondly, although we used Z-FA-FMK, a broad-spectrum cysteine protease inhibitor, it is important to note that the observed effects cannot be solely attributed to cathepsin B inhibition. The results likely reflect the inhibition of multiple cysteine proteases, with cathepsin B being a key player among them. To further validate cathepsin B's specific role, we also utilized AAV-mediated overexpression of cathepsin B and its inhibitor cystatin A, which supports our findings. Nevertheless, the use of more specific inhibitors or genetic approaches in future studies could help delineate the individual roles of these proteases. Despite these limitations, our findings with chemical and viral injury models, reveal PAJs as a key entry point into fibrosis, and provide a new direction to combat fibrotic diseases."

I still don't grasp the concept of transferred ECM, and my relative question still stand: How is it possible for such a highly complex and linked macromolecular structure to move around? By which mechanisms?

Response: The movement of ECM is an active, energy-dependent process. In the context of pulmonary fibrosis, alveolar macrophages and the mesothelium at pleural adhesion junctions are key players in this process. Specifically, Cathepsin B, which is activated by alveolar macrophages, degrades connective tissue at the pleural adhesion junctions, leading to the release of ECM macromolecules into lung interstitial spaces. These ECM macromolecules then migrate into the lung interior, where they contribute to the maintenance and progression of fibrosis.

Once the Cathepsin B signaling cascade has been activated in the mesothelium, the macrophages are no longer needed. Importantly, we found that inhibiting ECM transfer at PAJs through mesothelium-specific overexpression of Cystatin A can prevent downstream fibroblast activation and subsequent matrix deposition by interstitial fibroblasts. This suggests that targeting the disassembly of PAJs via the pleura may offer a novel therapeutic approach for resolving and potentially reversing lung fibrotic diseases.

Our previous research on skin and abdominal/peritoneal fibrosis was the first to report the phenomenon of ECM macromolecule transfer during fibrosis. These studies indicate that ECM transfer is a general feature of tissue and organ healing, where ECM from the serosal lining plays a crucial role in fibrotic repair. In the current study, we demonstrate that this mechanism is also active in the lungs, with ECM transfer and fibroblast activation together contributing significantly to the fibrotic material observed. While our earlier work suggested that fibroblasts could transport ECM cargo within the skin fascia, the mechanisms in the lung involve different cellular and molecular players, further underscoring the novelty and significance of our findings.

Figure 7

Reviewer #3 (Remarks to the Author):

The authors answered all my questions sufficiently. I would like to congratulate the authors on this work. I would just recommend the deletion of the following words: p.4, l.82: "as well as M1 and M2 macrophages". The M1/M2 concept is outdated and should not be propagated.

Response: Thank you very much for your professional suggestions and for recognizing the quality of our work. We have removed the phrase "as well as M1 and M2 macrophages".

RESPONSE TO REVIEWERS' COMMENTS

Reviewer #1:

Response: Thank you very much for recognizing our work. In accordance with your suggestion, we have added the following statement to the discussion section to prevent any potential misunderstanding:

“Lastly, while we use the term pleuro-alveolar junctions (PAJs) to facilitate discussion, we acknowledge that specific markers and precise definitions for PAJs remain provisional, requiring further research to clarify their role and contributions in fibrosis progression.”

Once again, we appreciate your valuable feedback.

Reviewer #2 (Remarks to the Author):

I am afraid that I remain unconvinced.

PAJs remain poorly defined. No specific molecular markers or relevant publications have been included although requested.

Non-specific FHS leakage upon inflammation and fibrosis was not sufficiently ruled out. Controls from previous studies in healthy tissues are not and cannot be adequately convincing.

Response: Thank you very much for your feedback. We appreciate your concerns and have made the following additions and clarifications in our manuscript. We have added a discussion on the limitations of our study, specifically regarding the term "pleuro-alveolar junctions" (PAJs) and the NHS-ester labeling method. The following statement has been included in the discussion: "While our study provides valuable insights into ECM dynamics and protease involvement, we acknowledge certain limitations. Firstly, although we have extensively validated the NHS-ester labeling method for studying ECM in vivo, additional experiments such as Evans blue assays or bleomycin IV administration could further confirm the specificity of the observed matrix transfer. Secondly, although we used Z-FA-FMK, a broad-spectrum cysteine protease inhibitor, it is important to note that the observed effects cannot be solely attributed to cathepsin B inhibition. The results likely reflect the inhibition of multiple cysteine proteases, with cathepsin B being a key player among them. To further validate cathepsin B's specific role, we also utilized AAV-mediated overexpression of cathepsin B and its inhibitor cystatin A, which supports our findings. Nevertheless, the use of more specific inhibitors or genetic approaches in future studies could help delineate the individual roles of these

proteases. Lastly, while we use the term pleuro-alveolar junctions (PAJs) to facilitate discussion, we acknowledge that specific markers and precise definitions for PAJs remain provisional, requiring further research to clarify their role and contributions in fibrosis progression."

Imaging, that the paper relies upon, remains unconvincing. No further imaging-associated (meta)data have been included (e.g. orthogonal and k-curve analyses) as requested.

Response: In response to your request for additional imaging data, we have uploaded the metadata and requested intensity profile analysis to Zenodo (DOI: 10.5281/zenodo.14357585). We have performed the K-curve analyses to compare intensity levels between different groups and assess colocalization in the revised figures as requested.

Therefore, the concept of mobile ECM remains not sufficiently convincing to me. How is it possible for a huge macromolecular structure to break all integrin-mediated cell attachments and migrate across barriers? Yes, proteolysis (and phagocytosis for that matter) could help but still ... How does mobile ECM integrate with the existing interstitial ECM?

Response: We appreciate your concerns. In our previous research, we established that ECM macromolecule transfer is a key aspect of tissue repair processes, specifically in skin (Correa-Gallegos D et al. CD201+ fascia progenitors choreograph injury repair [published correction appears in Nature. 2024 ; Correa-Gallegos D et al. Patch repair of deep wounds by mobilized fascia. Nature. 2019]) and abdominal/peritoneal fibrosis (Fischer A et al. Neutrophils direct preexisting matrix to initiate repair in damaged tissues. Nat Immunol. 2022; Fischer A et al. Post-surgical adhesions are triggered by calcium-dependent membrane bridges between mesothelial surfaces. Nat Commun. 2020). Building on this foundation, our current study explores the dynamics of ECM movement in pulmonary fibrosis, highlighting the critical interplay between alveolar macrophages and the mesothelium at pleural adhesion junctions.

In this context, Cathepsin B, activated by alveolar macrophages, plays a pivotal role in degrading connective tissue at these junctions. This degradation process releases ECM macromolecules into the lung interstitial spaces, enabling their migration and contributing to the progression of fibrosis.

Once the Cathepsin B signaling cascade is initiated in the mesothelium, the ongoing presence of macrophages is no longer essential for ECM movement. Our findings indicate that inhibiting ECM transfer at the PAJs through mesothelium-specific overexpression of Cystatin A can significantly prevent the activation of fibroblasts and their subsequent matrix deposition. This suggests

a promising therapeutic direction that targets the disassembly of PAJs to address lung fibrosis.